# Secondary lymphoid organ fibroblastic reticular cells mediate trans-infection of HIV-1 via CD44-hyaluronan interactions

Tomoyuki Murakami [1], Jiwon Kim[2], Yi Li [2,3], Glenn Edward Green[4], Ariella Shikanov [2,3] & Akira Ono [1]

Fibroblastic reticular cells (FRCs) are stromal cells in secondary lymphoid organs, the major sites for HIV-1 infection of CD4$^+$ T cells. Although FRCs regulate T cell survival, proliferation, and migration, whether they play any role in HIV-1 spread has not been studied. Here, we show that FRCs enhance HIV-1 spread via trans-infection in which FRCs capture HIV-1 and facilitate infection of T cells that come into contact with FRCs. FRCs mediate trans-infection in both two- and three-dimensional culture systems and in a manner dependent on the virus producer cells. This producer cell dependence, which was also observed for virus spread in secondary lymphoid tissues ex vivo, is accounted for by CD44 incorporated into virus particles and hyaluronan bound to such CD44 molecules. This virus-associated hyaluronan interacts with CD44 expressed on FRCs, thereby promoting virus capture by FRCs. Overall, our results reveal a novel role for FRCs in promoting HIV-1 spread.

---

[1] Department of Microbiology and Immunology, University of Michigan Medical School, Ann Arbor, MI 48109, USA. [2] Department of Macromolecular Science and Engineering, University of Michigan, Ann Arbor, MI 48109, USA. [3] Department of Biomedical Engineering, University of Michigan, Ann Arbor, MI 48109, USA. [4] Department of Otolaryngology-Head and Neck Surgery, University of Michigan Medical School, Ann Arbor, MI 48109, USA. Correspondence and requests for materials should be addressed to A.O. (email: akiraono@umich.edu)

Secondary lymphoid organs (SLOs), including lymph nodes (LNs), play a central role in dissemination of HIV-1. In both SIV-infected rhesus macaques[1–6] and HIV-1-infected humans[7], a large number of infected CD4[+] T cells are detectable in SLOs in contrast with peripheral blood. Furthermore, in infected individuals, SLOs are likely to harbor latent viral reservoirs[8–11] and therefore may become early sites of productive infection in the event of latent virus reactivation[12–14].

In LNs, T cells reside mainly in a T cell zone in which they are in constant contact with stromal cells known as fibroblastic reticular cells (FRCs)[15]. FRCs make a sponge-like network, which is an essential part of the T cell zone architecture[16]. The networks interact with several immune cells including T cells and thereby facilitate cell–cell contacts among them[15]. FRCs also modulate T cell properties via production of soluble factors including cytokine interleukin-7 (IL-7) and chemokines CCL19 and CCL21. These factors regulate T cell survival, proliferation, and migration[16,17]. Notably, these soluble factors are also known to alter susceptibility of T cells to HIV-1 infection or regulate the state of latency[18–20].

Although T cell zones and FRC networks therein are progressively damaged over the course of HIV-1 infection in vivo, which is implicated in CD4[+] T cell depletion[21], at early stages of the infection SIV-infected T cells are detectable in T cell zones of LNs in rhesus macaques[3,6]. Moreover, follicular helper T (Tfh) cells, which constitute a persistent reservoir in SLO germinal centers in aviremic individuals[5,11,22], are susceptible to infection in T cell zones while they are still precursors[23]. Infection of Tfh cells in follicles[22,24] may still occur near FRCs, since FRCs are also present in follicular regions[25]. Therefore, it is quite conceivable that FRCs regulate HIV-1 spread and persistence in LN T cells through their structural role or release of soluble factors. However, whether FRCs actually play any role in HIV-1 spread has not been studied.

In this study, we found that FRCs enhance HIV-1 spread by mediating trans-infection in both two- and three-dimensional (2D and 3D) culture systems. Notably, the cell type HIV-1 particles originated from was a key determinant for the FRC-mediated trans-infection and for efficient virus spread in an ex vivo human tonsil explant culture. We identified CD44 as the host factor that accounts for the observed producer cell dependence of trans-infection. Furthermore, a glycosaminoglycan, hyaluronan (HA), bound to CD44 on virus particles was also required for trans-infection. Finally, we found that FRCs capture virus particles via interactions between the HA on virus particles and CD44 on FRCs. These findings reveal the presence of a novel trans-infection mechanism mediated by stromal cells in SLOs and suggest that the interaction of HA and CD44 could be a new target for anti-HIV therapeutic strategies.

## Results

### The FRC-mediated enhancement of HIV-1 spread.
To investigate whether FRCs actually play any role in HIV-1 spread, we used FRCs isolated from human inguinal LNs (lnFRCs), which is commercially available as human lymphatic fibroblasts, and FRCs isolated from tonsils (tFRCs) of healthy donors according to an established protocol[26]. We confirmed that lnFRCs obtained from the commercial source expressed podoplanin (PDPN) and IL-7 but not CD31 as expected for FRCs[27] (Fig. 1a).

To investigate the effect of lnFRCs on HIV-1 infection, we examined HIV-1 spread in T cell-lnFRC cocultures by p24 enzyme-linked immunosorbent assay (ELISA). A CD4[+] T cell line, A3.01, was infected with HIV-1$_{NL4-3}$ in the presence and absence of lnFRCs, which are derived from two different donors. HIV-1 did not spread among lnFRCs alone. Remarkably,

however, coculturing A3.01 T cells with lnFRCs significantly accelerated HIV-1 spread compared to cultures of A3.01 cells alone and A3.01-HeLa cocultures (Fig. 1b). These results demonstrate that FRCs enhance HIV-1 spread without themselves being productively infected.

It is conceivable that the interaction of T cells with the same FRC, which has been observed in T cell zones[15], increases the frequency of T cell–T cell contact and thereby cell-to-cell transmission of HIV-1. Alternatively, FRCs may enhance HIV-1 spread through secretion of soluble factors that modulate HIV-1 infection (see Introduction). To test these hypotheses, we analyzed virus spread using the transwell culture system in which two groups of cells can be cultured in the same well but separated by a filter membrane permeable only to soluble factors and viruses. A3.01 T cells were infected with HIV-1$_{NL4-3}$ and stained with a vital dye CMTMR, which allows us to distinguish them from target T cells in coculture. We analyzed virus spread to target cells and virus expression among donor cells by monitoring Gag expression in CMTMR-negative and -positive cells, respectively, using flow cytometry. When the CMTMR-stained donor T cells and unstained target T cells were cocultured, the presence of lnFRCs enhanced virus spread substantially (Fig. 1c (a), compare lnFRCs (+) and (−)). In contrast, when the T cells were separated from lnFRCs by the transwell membrane, we did not observe such enhancement (Fig. 1c (b)), indicating that soluble factors released from lnFRCs are unlikely to play a role in enhancement of virus spread among T cells. As expected, when donor T cells are separated from target T cells by the transwell membrane, virus spread was less efficient compared to cultures without the separation (compare Fig. 1c (c) and (d) with (a) and (b)), which is likely due to the lack of cell-to-cell transmission. Notably, however, lnFRCs promoted virus spread in target cells even under these conditions (Fig. 1c (c) and (d), compare lnFRCs (+) and (−)). We noticed that the contacts of donor cells with lnFRCs increased virus spread 3–4-fold among these donor cells (Supplementary Fig. 1), which may consequently contribute to the increase of virus spread in target cells. Therefore, we normalized the increase in virus infection in target cells by the increase in infected donor cells to evaluate the true lnFRC-mediated enhancement of virus spread to target cells. This revealed that the lnFRC-mediated enhancement of the virus spread is dependent on the contact between uninfected target cells and lnFRCs (Fig. 1d). Altogether, these results suggest that neither soluble factors secreted by FRCs nor contacts between donor and target cells are required for FRC-mediated enhancement of HIV-1 spread and that contacts between FRCs and uninfected target T cells play a key role in promoting HIV-1 spread.

### Trans-infection of HIV-1 mediated by FRCs.
Dendritic cells (DCs) capture HIV-1 particles and facilitate HIV-1 transmission via DC–T cell contacts in the absence of productive infection in DCs. This process is termed trans-infection[28–30]. Since lnFRCs enhance HIV-1 spread to T cells in a manner dependent on lnFRC–T cell contacts but without productive infection in lnFRCs, it is possible that lnFRCs mediate trans-infection. To address this possibility, we examined virus capture, which is an essential and first step for trans-infection. In virus capture assays, lnFRCs and HeLa cells were pulsed for 2 h with HIV-1$_{NL4-3}$ produced by transfection of HeLa cells with pNL4-3, and after extensive washing, cell-associated virus was measured by p24 ELISA. We found that lnFRCs captured HIV-1 more efficiently than HeLa cells (Fig. 1e). To investigate whether the virus-loaded lnFRCs mediate trans-infection, A3.01 T cells were added to the virus-pulsed lnFRCs and cocultured for 6 days. Gag expression in

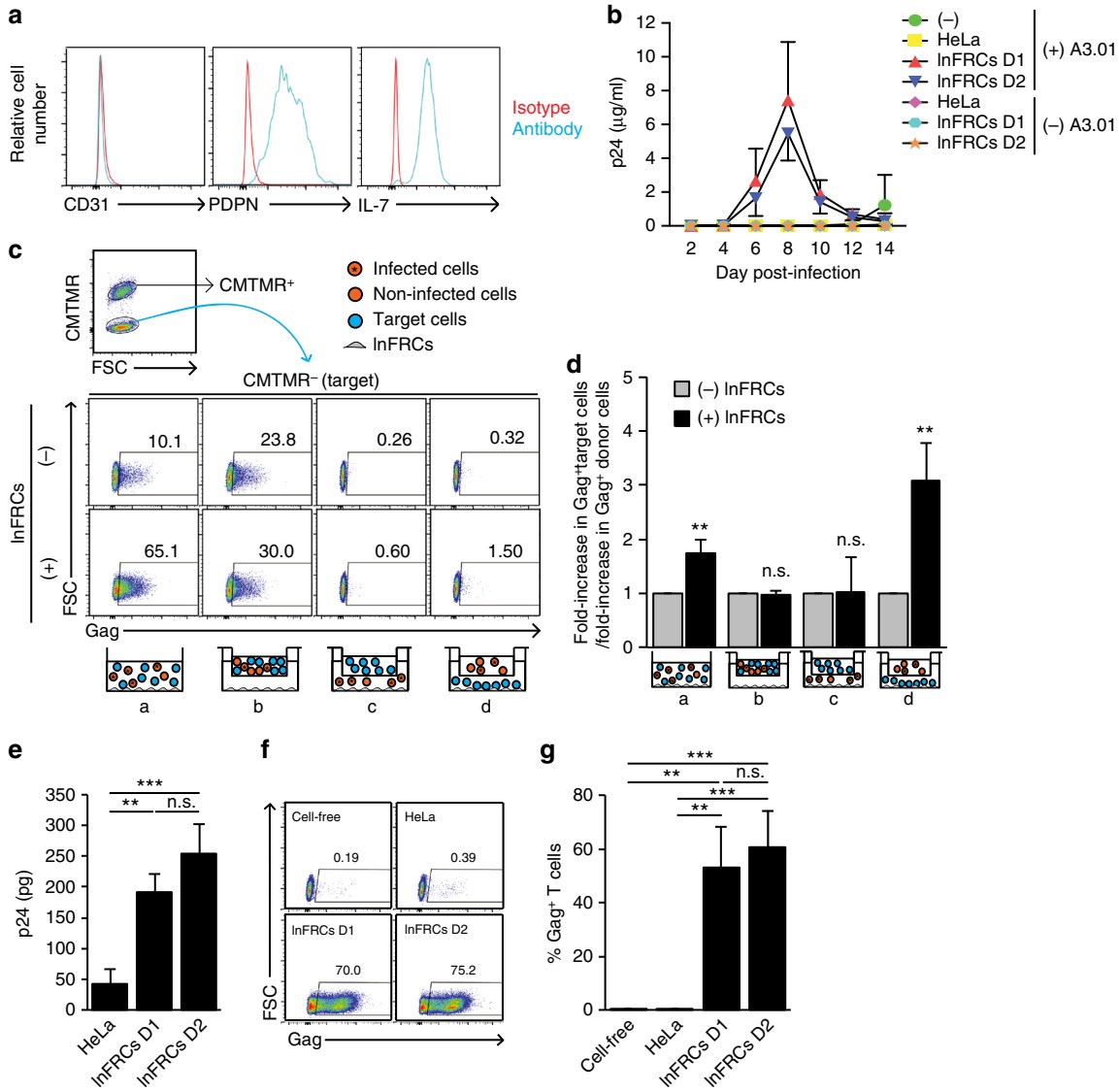

**Fig. 1** Lymph node FRCs enhance HIV-1 spread via trans-infection. **a** Flow cytometry analysis of FRC markers on lymph node FRC (lnFRC) surface. Similar results were obtained using lnFRCs isolated from three different donors. **b** A3.01 T cells were inoculated with 0.254 ng p24 of HIV-1$_{NL4-3}$ in the presence or absence of HeLa cells or lnFRCs in 1 ml RPMI-10. To analyze infection of lnFRCs, lnFRCs were also inoculated with the same amount of HIV-1$_{NL4-3}$ in the absence of A3.01 T cells. To analyze HIV replication kinetics in A3.01 T cells in the presence or absence of HeLa cells or lnFRCs, the 50-µl culture supernatants were collected every 2 days and examined using the p24 ELISA assay. After each collection of the 50-µl supernatants, the culture was gently resuspended, 700 µl of the cell suspension was discarded, and 750 µl of fresh RPMI-10 was added. During the experimental period, lnFRCs were not detached but kept adherent to the bottom of culture wells. lnFRCs isolated from two different donors (D1 and D2) were used. **c** Flow cytometry analysis of HIV-1 infection in the transwell system. Following coculture with CMTMR-stained HIV-1-infected A3.01 T cells in the presence or absence of lnFRCs and transwell inserts, CMTMR$^-$ A3.01 T cells were examined for expression of Gag as detailed in Methods. lnFRCs isolated from one donor were used. **d** Fold-increases in Gag$^+$ cell number in CMTMR$^-$ cells (an example raw data shown in **c**) were normalized by fold-increases in Gag$^+$ cells in CMTMR$^+$ cells (Supplementary Fig. 1). **e** The virus capture assay. Following 2-h incubation with HIV-1$_{NL4-3}$ (7.6 ng p24) and washing with PBS, the amount of virus captured by HeLa cells and lnFRCs isolated from two different donors was quantified in p24 ELISA assays. **f, g** The trans-infection assay. An example of flow cytometry analysis of A3.01 T cells infected via trans-infection mediated by HeLa cells or lnFRCs isolated from two different donors is shown in **f**. Data shown in **g** was calculated from three independent experiments performed as in **f**. HeLa cells and lnFRCs were cultured in the presence of HIV-1$_{NL4-3}$ (7.6 ng p24) for 2 h, washed extensively, and cocultured with A3.01 T cells for 6 days. For the cell-free condition, A3.01 T cells were inoculated with the same amount of input virus and cultured for 6 days without removing the inoculum. A3.01 T cells were analyzed for expression of Gag. Data shown in **b**, **d**, **e**, and **g** represent the mean ± SD of three independent experiments. The $p$-values were determined using two-tailed Student's $t$-test for **d** and Tukey's test following one-way ANOVA for **e** and **g**. **$p < 0.01$; n.s., not significant

A3.01 T cells was analyzed by flow cytometry. In contrast to when A3.01 T cells were inoculated with the same amount of input virus in the absence of lnFRCs (cell-free), we observed a robust infection when T cells were cocultured with virus-pulsed lnFRCs (Fig. 1f, g), indicating that lnFRCs mediate trans-infection.

To compare the efficiency of trans-infection mediated by lnFRCs with that of immature DC (imDC)- and mature DC (mDC)-mediated trans-infection, imDCs and mDCs were generated from monocytes (Supplementary Fig. 2A). lnFRCs, imDCs, and mDCs were pulsed for 2 h with HeLa-derived HIV-

$1_{NL4-3}$, and after washing, A3.01 T cells were cocultured with the virus-pulsed cells for 6 days. We found that at least under the

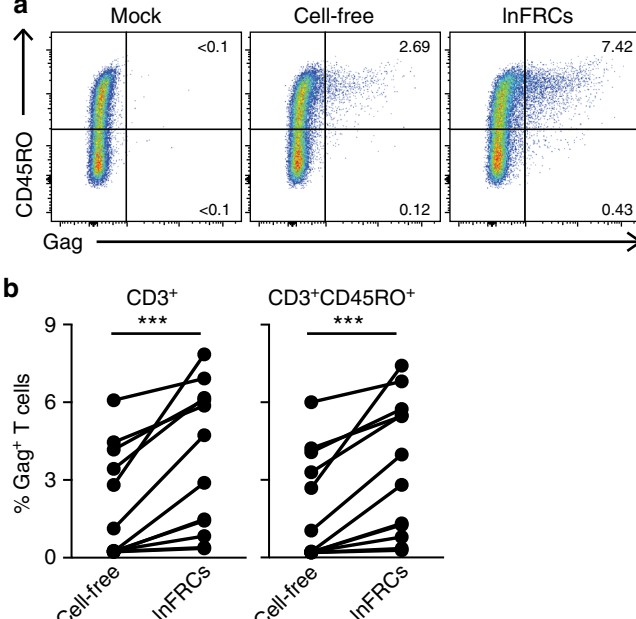

**Fig. 2** FRCs mediate trans-infection of memory CD4$^+$ T cells. lnFRCs were cultured in the presence of HIV-1$_{NL4-3}$ (100 ng p24) for 2 h, washed extensively, and cocultured with PHA-stimulated peripheral blood leukocytes (PBLs) for 4 days. For the cell-free condition, PHA-stimulated PBLs were inoculated with the same amount of input virus and cultured for 4 days without washing off the inoculum. Gag expression among CD3$^+$CD45RO$^+$ and CD3$^+$CD45RO$^-$ cells were analyzed by flow cytometry. **a** A representative set of the results obtained using one pair of lnFRCs and PBLs is shown. **b** Data represent the results of experiments performed using 12 different pairs of lnFRCs and PBLs. lnFRCs derived from four different donors were paired with PBLs isolated from one of eleven different donors. The *p*-values were determined using Wilcoxon matched-pairs signed rank test. ***$p < 0.001$

condition used, lnFRCs mediated trans-infection of A3.01 T cells more efficiently than imDCs and mDCs (Supplementary Fig. 2B).

A major target of HIV-1 is CD4$^+$ memory T cells in vivo[31,32]. To investigate whether FRCs mediate trans-infection of primary memory CD4$^+$ T cells, we used PHA-stimulated peripheral blood lymphocytes (PBLs) as target cells. In these experiments, we increased the inoculum to 100 ng p24 HIV-1$_{NL4-3}$, because compared to A3.01 T cells, primary T cells are generally less susceptible to HIV-1 infection. lnFRCs were incubated with the virus for 2 h, washed extensively, and cocultured with PBLs for 4 days. Gag expression in CD3$^+$CD45RO$^+$ memory T cells and CD3$^+$CD45RO$^-$ T cells was analyzed by flow cytometry. CD3$^+$ cells expressing Gag increased significantly when PBLs were cocultured with virus-pulsed lnFRCs relative to cell-free infection (Fig. 2a, b). It is worth noting that in this comparison, 100 ng virus (in p24) remained in cell-free infection cultures for 4 days, whereas the amount of input virus that remains after washing in the trans-infection cultures (i.e., virus captured by lnFRCs) was typically ~2 ng (see Dependence of trans-infection on virus producer cell types). Therefore, there was ~50-fold more virus present in cell-free cultures than in trans-infection cultures initially. Nonetheless, PBL infection was enhanced in the presence of lnFRCs relative to cell-free infection. When PBLs in cell-free cultures were washed after the 2-h inoculation period to remove unbound virus as done with lnFRCs in the coculture conditions,

we observed on average 6.9-fold increase in Gag$^+$ cells in the presence of lnFRCs compared to cell-free cultures (Supplementary Fig. 3A and B). Gag$^+$ cells were predominantly CD45RO$^+$ memory T cells in both cell-free PBL infection cultures and PBL-lnFRC cocultures, and the enhancement of infection in the cocultures relative to cell-free infection cultures was observed in the CD45RO$^+$ memory T cell population (Fig. 2a, b; Supplementary Fig. 3A and B). These results indicate that FRCs mediate trans-infection of memory CD4$^+$ T cells. We noticed a variation in the extent of lnFRC-mediated enhancement of infection among PBLs isolated from different donors. Therefore, in the subsequent studies, we used A3.01 T cells as target cells when we need to verify that virus captured by FRCs are transmittable to T cells.

**Trans-infection of HIV-1 produced by SLO-associated T cells.** To determine whether FRCs mediate trans-infection of HIV-1 released from SLO-associated T cells, we propagated HIV-1$_{NL4-3}$ in human lymphoid aggregate cultures (HLACs) prepared from tonsils and used the virus as an inoculum. We found that lnFRCs mediate trans-infection of HLAC-derived virus (Fig. 3a). In addition, we observed that tFRCs mediate trans-infection of virus derived from HLACs of the same tonsil donor in A3.01 T cells (Fig. 3b and Supplementary Fig. 4), indicating that the ability to mediate trans-infection is not limited to FRCs from inguinal LNs. Altogether, results shown in Figs. 1–3 demonstrate that HIV-1 captured by FRCs is accessible to CD4$^+$ T cells susceptible to infection and that FRCs mediate trans-infection of HIV-1, including virus produced by T cells present in SLOs.

**Dependence of trans-infection on virus producer cell types.** DC-mediated trans-infection relies on DC-SIGN, which recognizes the viral envelope glycoprotein gp120 (Env), and CD169, which interacts with gangliosides incorporated into HIV-1 particles[28–30]. CD169 was also recently shown to promote trans-infection mediated by macrophages lining the subcapsular sinus in LNs[33]. To test whether these molecules are involved in trans-infection mediated by lnFRCs, we analyzed DC-SIGN and CD169 expression in lnFRCs. We found that lnFRCs do not express either protein (Fig. 4a, b). These results suggest that the molecular mechanism of trans-infection mediated by FRCs is different from that by imDCs and mDCs.

In addition, Env, the only viral protein that could interact with FRC surface molecules, was not required for virus capture by lnFRCs (Fig. 4c), indicating that virus-associated host molecules are involved in the FRC-mediated trans-infection.

The host molecule profile on HIV-1 particles varies depending on producer cell types[34]. Thus, to identify the virus-associated molecule that is important for FRC-mediated trans-infection, we sought to compare virus stocks produced by different producer cells in the virus capture and trans-infection assays. To our surprise, virus stocks prepared using HeLa and 293T cells showed a striking difference. We found that when the same amount of input HIV-1$_{NL4-3}$ (based on the p24 amount) was used, lnFRCs captured HeLa-derived virus more efficiently than 293T-derived virus (left 2 panels in Fig. 5a). The difference in captured amounts of HeLa- versus 293T-derived viruses was observed regardless of the amounts of input virus (7.6 and 100 ng p24; Fig. 5a). tFRCs also showed similar virus-producer-cell dependency in virus capture although the difference was somewhat smaller (right panel in Fig. 5a). Notably, nine primary transmitted/founder HIV-1 clones as well as a chimeric infectious HIV-1 subtype C molecular clone MJ4, which originates from later-phase patients[35], were efficiently captured by lnFRCs when the viruses were generated using HeLa cells but not 293T cells (Fig. 5b and

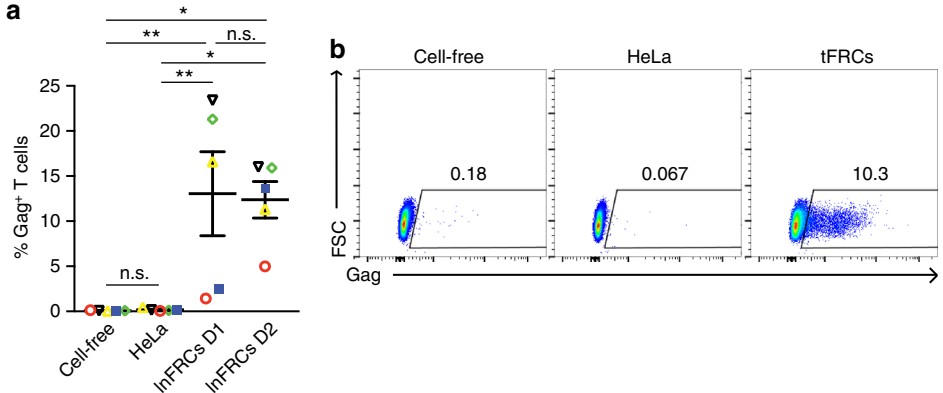

**Fig. 3** FRCs mediate trans-infection of HIV-1 produced by HLACs. The efficiencies of trans-infection of human lymphoid aggregate culture (HLAC)-derived virus mediated by HeLa cells, lnFRCs or tonsillar FRCs (tFRCs) were compared using flow cytometry. HeLa cells and lnFRCs or tFRCs were cultured in the presence of HIV-1$_{NL4-3}$ (25.4 ng p24) for 2 h, washed extensively, and cocultured with A3.01 T cells for 6 days. For the cell-free condition, A3.01 T cells were inoculated with the same amount of input virus and cultured for 6 days without removing the initial inoculum. **a** HLACs were prepared from the tonsils of five different donors. Data represent the results of experiments performed using five different virus stocks produced in these HLACs (each symbol corresponds to a virus stock prepared using HLACs of one unique donor). lnFRCs from two donors were used. **b** tFRCs and HLACs were prepared from the tonsils of the same donor in each experiment as detailed in Methods. Trans-infection of HIV-1 propagated in HLACs by tFRCs isolated from the same tonsil donor was examined. A representative set of the results obtained using tFRCs and HLACs from one donor is shown. The data obtained with tonsillar cells from all five different donors are shown in Supplementary Fig. 2. The mean ± SEM is shown. The *p*-values were determined using Tukey's test following one-way ANOVA. *$p < 0.05$; **$p < 0.01$; n.s., not significant

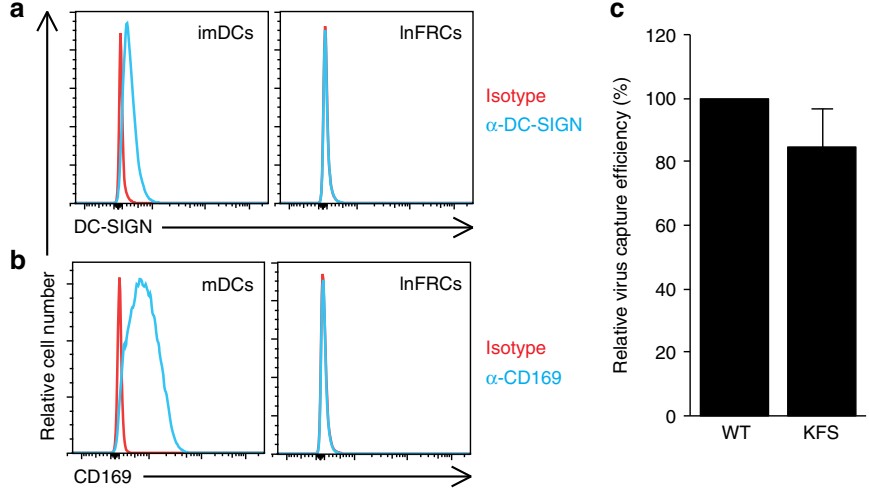

**Fig. 4** FRCs mediate trans-infection in a DC-SIGN- and CD169-independent mechanism. **a** Flow cytometry analysis of DC-SIGN expression on the surface of lnFRC and immature DC (imDC), the latter of which serves as a positive control. **b** Flow cytometry analysis of CD169 expression on the surface of lnFRC and mature DC (mDC), the latter of which serves as a positive control. **c** Quantification of the amount of HeLa-derived WT and KFS (Env⁻) HIV-1$_{NL4-3}$ captured by lnFRCs. lnFRCs were cultured in the presence of HIV-1$_{NL4-3}$ (7.6 ng p24) for 2 h and washed extensively. The amount of virus captured by lnFRCs was quantified by p24 ELISA assays. Similar results were obtained using lnFRCs isolated from three different donors. Data represent the mean ± SD of three independent experiments

Supplementary Fig. 5A). These results indicate that FRCs capture not only a lab-adapted strain but also primary isolates in a manner dependent on virus producer cells. Consistent with the virus capture outcomes, lnFRC-dependent trans-infection of A3.01 T cells was significantly more efficient with HeLa-derived virus than with 293T-derived virus (Fig. 5c). These results suggest a possibility that HIV-1 incorporates molecule(s) important for FRC-mediated trans-infection, which is differentially expressed between HeLa and 293T cells.

**HIV-1 spread in 3D culture systems that contain FRCs.** In a T cell zone of LNs in vivo, FRCs form a reticular network[15], which

can be recapitulated in a 3D culture[36]. Mechanical processes that drive T cell migration differ in 2D and 3D environments[37,38]. In addition, formation of motile HIV-infected T cell syncytia, which have been observed in LNs in a mouse model and hypothesized to contribute to virus spread in vivo, was detected in a 3D culture system, but not in conventional suspension cultures[39,40]. Thus, it appears important to test whether FRCs capture virus and mediate trans-infection in a 3D condition that resembles the LN T cell zone. To examine the ability of FRCs forming such networks to mediate trans-infection, we established a 3D culture system. Immortalized lnFRCs (ilnFRCs), which are PDPN⁺ CD31⁻ and mediate trans-infection of A3.01 T cells in 2D conditions like parental primary lnFRCs (Supplementary Fig. 5B–D),

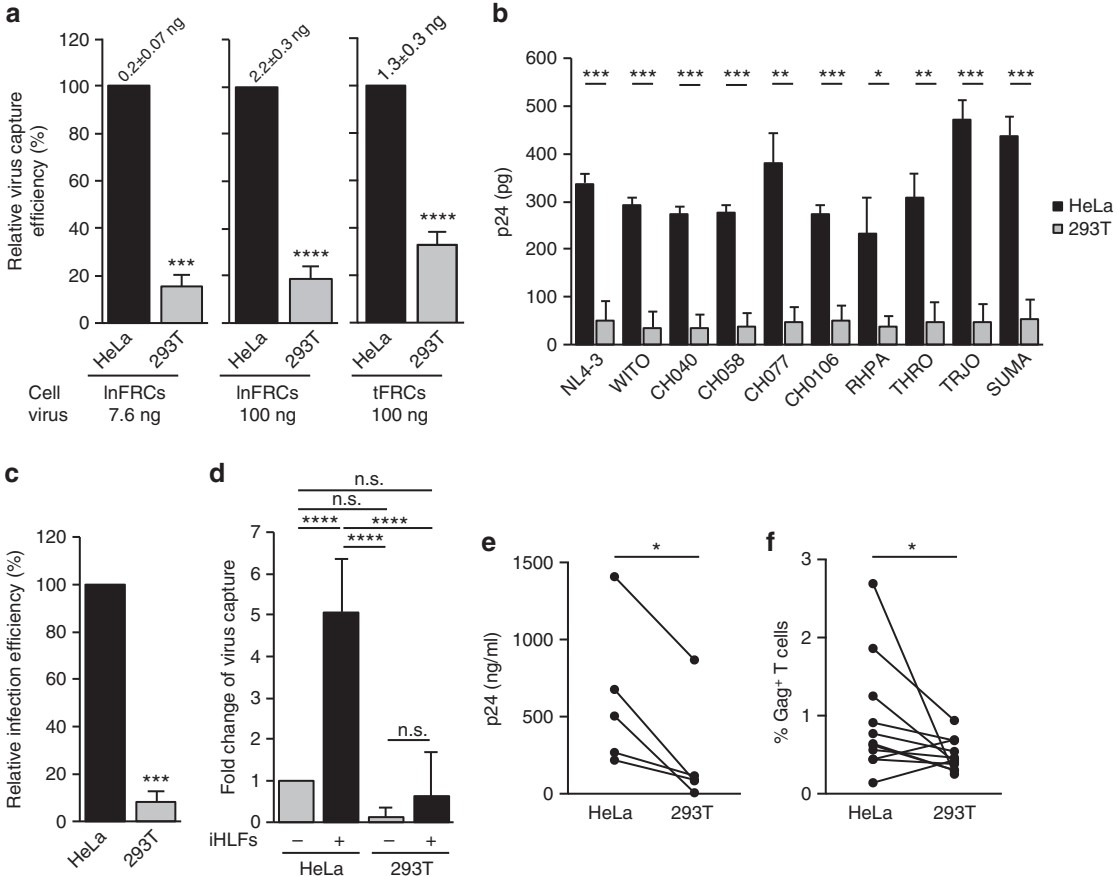

**Fig. 5** FRCs mediate trans-infection in two-dimensional and three-dimensional culture systems in a manner dependent on virus producer cell types. The input viruses were produced using HeLa and 293T cells and compared using the same amount (based on the p24 amount). **a** Quantification of the amount of HIV-1$_{NL4-3}$ captured by lnFRCs and tFRCs. lnFRCs and tFRCs were cultured in the presence of the indicated amount of HIV-1$_{NL4-3}$ for 2 h and washed extensively. The amount of virus captured by lnFRCs and tFRCs was quantified by p24 ELISA assays. Relative virus capture efficiency is calculated in comparison with the amount of HeLa-derived virus captured by lnFRCs or tFRCs in each experiment. Absolute amounts of captured HeLa-derived virus are shown above the black bars. **b** Quantification of the amount of transmitted/founder viruses captured by lnFRCs. For inoculation, 7.6 ng p24 of viruses were used. **c** Comparison of efficiencies of trans-infection of A3.01 T cells mediated by lnFRCs. Performed as in Fig. 1. **d** Comparison of virus capture efficiencies in a 3D culture system encapsulating ilnFRCs. PEG gels encapsulating either ilnFRCs or no cells were pulsed with HeLa- and 293T-derived HIV-1$_{NL4-3}$ (7.6 ng p24) for 16 h and washed extensively. The gels were further cultured for 4 h and washed with RPMI-10 twice. The gels were solubilized, and the amount of virus in solubilized gels was quantified by p24 ELISA assays. Fold changes relative to the level of HeLa-derived virus captured in the PEG gel that does not contain ilnFRCs are shown. **e** A3.01 T cells were added to the 3D culture system pulsed with HeLa- and 293T-derived viruses (7.6 ng p24) and cultured for 6 days. Virus production following trans-infection was compared based on p24 concentrations in supernatants. Data represent the results of five independent experiments. **f** Percentages of infected CD3$^+$ T cells in tonsil blocks inoculated with HeLa- and 293T-derived viruses (25.4 ng p24) were compared ($n = 11$ donors). Tonsil blocks were inoculated with HIV-1 and cultured for 6 days. The blocks were digested to obtain cell suspension, and CD3$^+$ T cells were examined for expression of Gag by flow cytometry. Data shown in **a–d** represent the mean ± SD of three independent experiments except for tFRC experiments shown in **a**. For lnFRC experiments shown in **a–c**, similar results were obtained with lnFRCs isolated from three different donors. For tFRC experiments in **a**, the mean ± SD is shown for tFRCs from three different donors. The $p$-values were determined using two-tailed Student's $t$-test for **a–c**, Tukey's test following one-way ANOVA for **d**, two-tailed paired Student's $t$-test for **e** and Wilcoxon matched-pairs signed rank test for **f**. *$p < 0.05$; **$p < 0.01$; ***$p < 0.001$; ****$p < 0.0001$; n.s., not significant

were encapsulated in a PEG gel and allowed to grow into a 3D network (Supplementary Fig. 5E). In this system, A3.01 T cells were capable of forming contacts with the ilnFRC networks (Supplementary Fig. 5F). Using this system, we found that under the 3D condition, ilnFRCs capture, and facilitate spread of, HeLa-derived virus more efficiently than 293T-derived virus among A3.01 T cells (Fig. 5d, e). Consistent with the role of FRCs in trans-infection in the 3D context of SLOs, HeLa-derived virus efficiently disseminated in an ex vivo human tonsil culture system compared to 293T-derived virus (Fig. 5f), even though the relative infectivity of 293T-derived virus used in these experiments was higher than that of HeLa-derived virus (Supplementary Fig. 5G). Altogether, these results revealed that the networks of FRCs

mediate trans-infection of HIV-1 in a manner dependent on virus producer cell types and strongly suggest that this process facilitates HIV-1 spread in the 3D context of SLOs.

**Dependence of trans-infection on CD44 in virions**. We recently reported that T cell uropod proteins, CD43, CD44, and PSGL-1, cocluster with Gag on the cell surface of virus-expressing cells, which likely promotes incorporation of these proteins into virus particles[41]. These molecules are involved in cell–cell contacts[42], suggesting a possible role for these uropod proteins in the interaction of virus with FRCs. Among them, CD44 showed a higher expression in HeLa cells than 293T cells (Fig. 6a), whereas

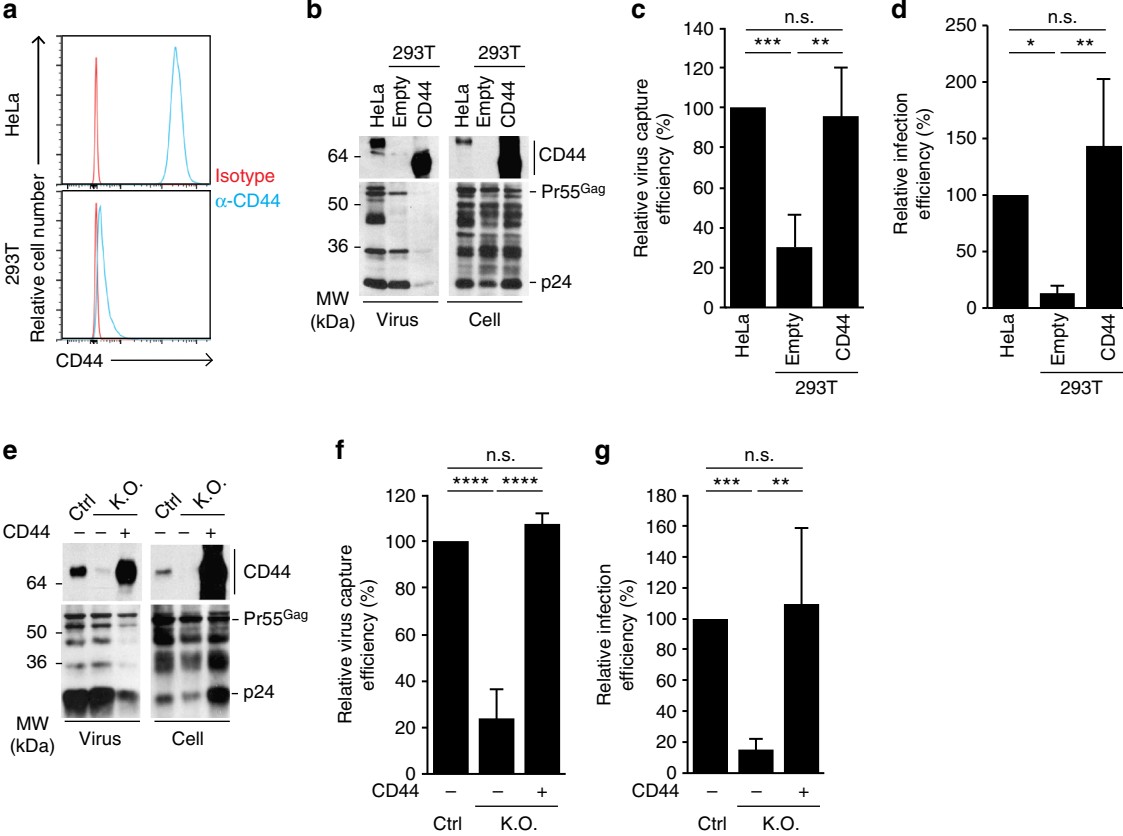

**Fig. 6** The incorporation of CD44 into virions is required for efficient FRC-mediated trans-infection. **a** Flow cytometry analysis of CD44 expression on HeLa and 293T surfaces. **b** Western blot analysis of CD44 incorporation into virus particles, which were produced by HeLa and 293T cells, latter of which had been transfected with empty or CD44-encoding expression plasmids. CD44 (top) and HIV-1 proteins (bottom) are shown for virus (left) and cell (right) lysates. **c** HIV-1$_{NL4-3}$ generated as in **b** was compared for efficiencies of virus capture by lnFRCs. lnFRCs were cultured in the presence of HIV-1$_{NL4-3}$ (7.6 ng p24) for 2 h and washed extensively. The amount of virus captured by lnFRCs was quantified by p24 ELISA assays. Relative efficiencies in comparison to HeLa-derived virus are shown. **d** The viruses examined in **c** were compared for the efficiencies of lnFRC-mediated trans-infection of A3.01 T cells. lnFRCs were cultured in the presence of HIV-1$_{NL4-3}$ (7.6 ng p24) for 2 h, washed extensively, and cocultured with A3.01 T cells for 6 days. A3.01 T cells were analyzed for expression of Gag by flow cytometry. **e** Western blot analysis of CD44 incorporation into virus particles, which were produced by control (Ctrl), CD44-knocked out (K.O.), and CD44-reconstituted HeLa cells. CD44-knocked out HeLa cells were generated using a CRISPR-Cas9-based method. Control HeLa cells were prepared similarly except that pSpCas9(BB)-2A-Puro was used instead of pSpCas9(BB)-2A-Puro/CD44-gRNA. Reconstitution was achieved via transfection of the CD44 expression plasmid (+). Virus and cell lysates were analyzed as in **b**. **f** HIV-1$_{NL4-3}$ generated as in **e** was compared for efficiencies of virus capture by lnFRCs. Virus capture was analyzed as in **c**. **g** The viruses examined in **f** were compared for the efficiencies of lnFRC-mediated trans-infection of A3.01 T cells. Trans-infection was analyzed as in **d**. For experiments in **c**, **d**, **f**, and **g**, the data represent the mean ± SD of three (**f**) and four (**c**, **d**, and **g**) independent experiments. These experiments were performed using lnFRCs from one donor, but we observed similar results by using lnFRCs isolated from two other donors (total three donors). The p-values were determined using Tukey's test following one-way ANOVA. *$p < 0.05$; **$p < 0.01$; ***$p < 0.001$; ****$p < 0.0001$; n.s., not significant

no difference was observed for CD43 and PSGL-1 (Supplementary Fig. 6A). To investigate the effect of CD44 incorporation on virus capture and trans-infection, we co-transfected an expression plasmid for CD44 and pNL4-3 in 293T cells. The molecular weight of overexpressed CD44 in 293T cells was different from CD44 in HeLa cells, probably due to differential glycosylation (Fig. 6b). Nevertheless, ectopically expressed CD44 was incorporated into released virus particles (Fig. 6b and Supplementary Fig. 6B). Importantly, the ectopic expression of CD44 enhanced lnFRC-mediated capture and trans-infection to A3.01 T cells of virus produced by 293T cells (Fig. 6c, d). We further tested the effect of knocking out endogenous CD44 in virus producer HeLa cells using a CRISPR/Cas9 approach. After transfection of pSpCas9(BB)-2A-Puro/CD44-gRNA and selection using puromycin, we obtained HeLa cells that do not express detectable CD44 (Supplementary Fig. 6C). As expected, CD44 incorporated into the virus produced in these cells was nearly undetectable (Fig. 6e). Importantly, the virus produced by CD44-knocked out

cells failed to undergo lnFRC-mediated capture and trans-infection of A3.01 T cells. However, overexpression of CD44 in these CD44-knocked out cells restored CD44 incorporation into virions as well as lnFRC-mediated virus capture and trans-infection (Fig. 6e–g). These results demonstrate that CD44 incorporated into virus particles plays an important role in trans-infection mediated by FRCs.

**Trans-infection mediated by CD44–hyaluronan interactions**. A main ligand of CD44 is a glycosaminoglycan HA[43]. Since HA was detected on the surface of lnFRCs (Supplementary Fig. 6D), it is conceivable that CD44 incorporated in virus particles binds to HA on the surface of FRCs, thereby promoting virus capture by FRCs. Alternatively, since CD4+ T cells as well as HeLa cells express HA on their surface[44–46], such HA may bind to CD44 on nascent virus particles and further mediate binding of the particles to an unoccupied HA receptor expressed on FRCs. Consistent with this possibility, exogenous HA reduced lnFRC-mediated virus capture and

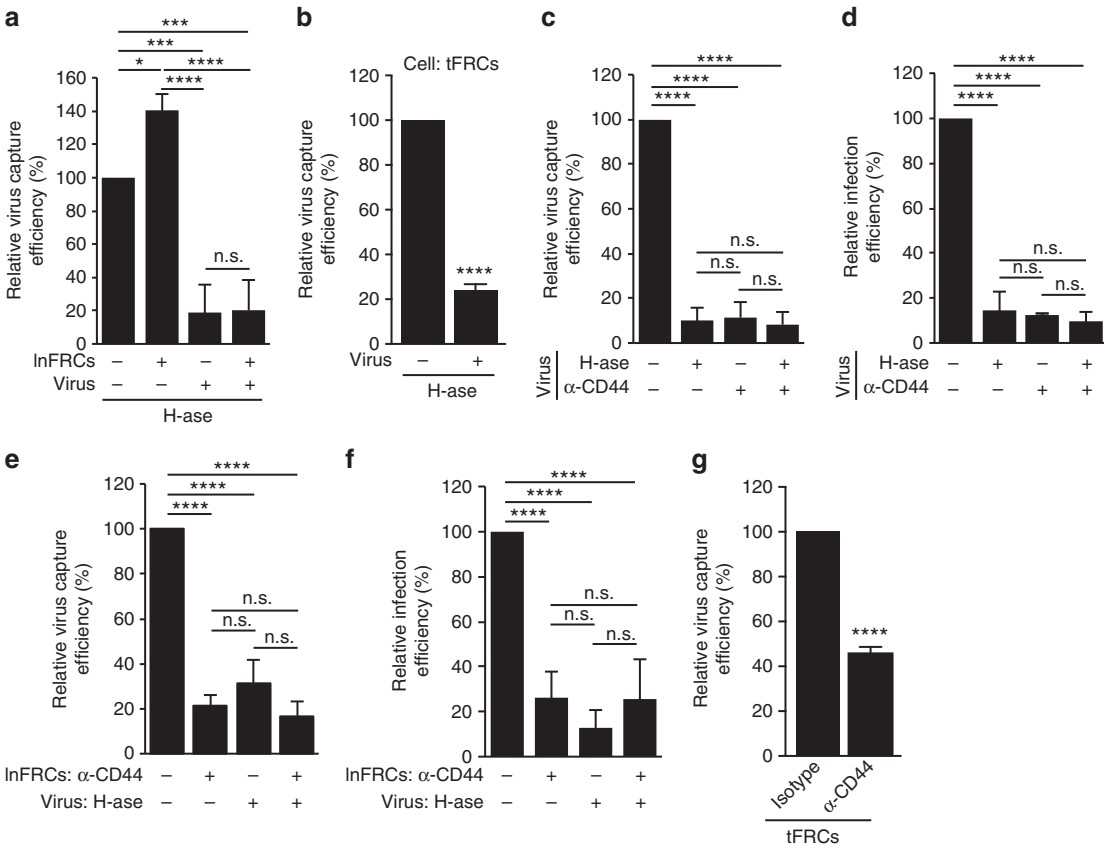

**Fig. 7** Interactions between CD44 and HA promote FRC-mediated trans-infection. **a** HIV-1 generated in HeLa cells was either untreated or pretreated with 10 units/ml hyaluronidase (H-ase) for 1 h and used as an inoculum without separating virus from H-ase. lnFRCs were either untreated or pretreated with 10 units/ml H-ase for 1 h, and then H-ase was removed by washing with PBS twice prior to addition of the untreated or H-ase-treated virus inoculums. The amount of virus captured by lnFRCs was quantified by p24 ELISA assays. **b** tFRCs were cultured in the presence of HeLa-derived HIV-1$_{NL4-3}$ (100 ng p24), which were either untreated or pretreated with 25 units/ml H-ase, for 1 h. After 2-h incubation with the virus, cells were washed extensively, and the amount of virus captured by tFRCs was quantified by p24 ELISA assays. **c, d** HeLa-derived HIV-1 was pretreated with 10 units/ml H-ase and/or 0.625 µg/ml anti-CD44 antibody for 1 h and used as inoculums without separating virus from H-ase or anti-CD44. lnFRC-mediated virus capture and trans-infection to A3.01 T cells were examined as in Fig. 6c, d, respectively. **e, f** HeLa-derived HIV-1 was either untreated or pretreated with 10 units/ml H-ase for 1 h and used as inoculums. lnFRCs were either untreated or pretreated with 10 µg/ml anti-CD44 antibody for 1 h, and excess antibodies were removed by washing with PBS twice prior to addition of the untreated or H-ase-treated virus inoculums. lnFRC-mediated capture of viruses and trans-infection to A3.01 T cells were examined as in Figure 6c, d, respectively. **g** tFRCs were pretreated with 10 µg/ml anti-CD44 antibody for 1 h, and excess antibodies were removed by washing with PBS twice prior to addition of HeLa-derived HIV-1$_{NL4-3}$ (100 ng p24). Virus capture was examined as in **b**. For experiments in **a**, **c–f**, the data represent the mean ± SD of three (**a**, **c–e**) and four (**f**) independent experiments. These experiments were performed using lnFRCs from one donor, but we observed similar results by using lnFRCs isolated from two other donors (total three donors). For experiments shown in **b** and **g**, tFRCs from three different donors were used. The p-values were determined using Tukey's test following one-way ANOVA for **a**, **c–f** and two-tailed Student's t-test for **b** and **g**. *p < 0.05; ***p < 0.001; ****p < 0.0001; n.s., not significant

trans-infection of A3.01 T cells (Supplementary Fig. 6E and F). To test these hypotheses, we examined the effects of hyaluronidase (H-ase). lnFRCs were either pretreated with H-ase or not, washed to remove H-ase, and incubated with either H-ase-treated or untreated HeLa-derived virus. Pretreatment of lnFRCs with H-ase enhanced virus capture by lnFRCs (Fig. 7a). By contrast, pretreatment of virus with H-ase suppressed virus capture by lnFRCs, regardless of whether lnFRCs were also pretreated with H-ase (Fig. 7a). Likewise, we found that H-ase pretreatment of HeLa-derived virus diminishes virus capture by tFRCs (Fig. 7b). Furthermore, we observed that pretreatment of virus with the anti-CD44 antibody, H-ase, or both causes similar reduction in lnFRC-mediated virus capture and trans-infection of A3.01 T cells (Fig. 7c, d). These results indicate that HA bound to CD44 on virus surface but not HA on the surface of FRCs is required for FRC-mediated trans-infection.

Treatment of lnFRCs with H-ase enhanced capture of HeLa-derived virus (Fig. 7a), suggesting that HA preexisting on lnFRCs

may compete with HA associated with virus particles for binding to an HA receptor on the lnFRC surface such as CD44. We observed that lnFRCs express CD44 on the cell surface (Supplementary Fig. 6G) although its size was apparently different from the size of CD44 expressed in HeLa cells (Supplementary Fig. 6H). In addition, CD44 was also expressed on the tFRC surface (Supplementary Fig. 6I). To examine the role of CD44 expressed on FRCs more directly, lnFRCs were pretreated with the anti-CD44 antibody that blocks HA binding[47] prior to addition of HeLa-derived virus. The pretreatment reduced virus capture and trans-infection of A3.01 T cells to a similar extent as the treatment of HeLa-derived virus with H-ase (Fig. 7e, f). Albeit modest, virus capture by tFRCs also showed sensitivity to the pretreatment with the same anti-CD44 antibody (Fig. 7g).

Since it is possible that the binding of a monoclonal antibody to cell surface CD44 suppress virus capture via steric hindrance rather than the inhibition of binding of CD44 with HA, we tested

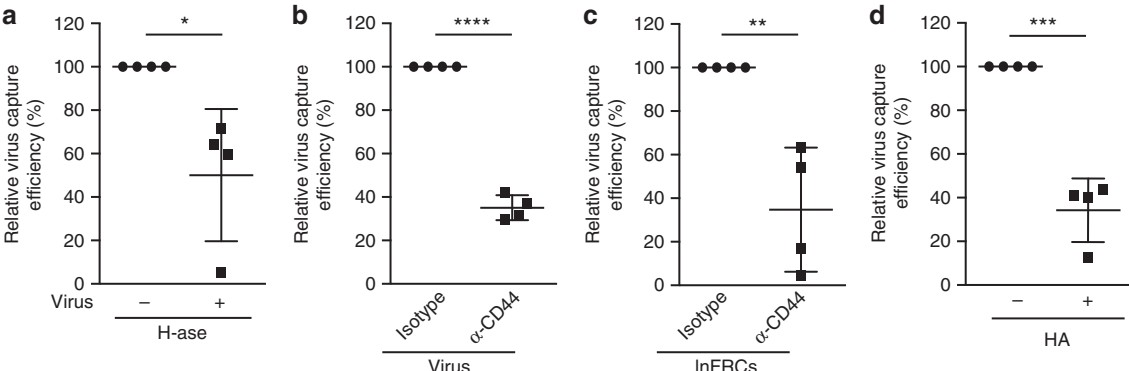

**Fig. 8** HLAC-derived virus is captured by lnFRCs in a HA- and CD44-dependent manner. The effects of the following treatments on virus capture by lnFRCs were examined using 25.4 ng p24 of HLAC-derived virus as inoculums. The amount of captured virus was quantified by p24 ELISA assays. $n = 4$ different HLAC donors. **a** HLAC-derived virus was either untreated or pretreated with 10 units/ml hyaluronidase (H-ase) for 1 h. lnFRCs were incubated with the virus for 2 h without separating virus from H-ase. **b** HLAC-derived virus was either untreated or pretreated with 0.625 μg/ml anti-CD44 for 1 h. lnFRCs were incubated with the virus for 2 h without separating virus from anti-CD44. **c** lnFRCs were either untreated or pre-incubated with 10 μg/ml anti-CD44 antibody for 1 h, and excess antibodies were removed by washing cells with PBS twice. The cells were then incubated with HLAC-derived virus for 2 h. **d** lnFRCs were incubated with HLAC-derived virus in the absence or presence of exogenous <10 kDa HA (2.5 mg/ml) for 2 h. lnFRCs from one donor were used for these experiments. Data represent the mean ± SD. The p-values were determined using two-tailed Student's t-test. *p < 0.05; **p < 0.01;***p < 0.001; ****p < 0.0001

whether another monoclonal antibody that binds CD44 via a region unrelated to HA binding inhibits capture of HeLa-derived virus by lnFRCs. We used an anti-CD44 monoclonal antibody (clone L178), which binds to CD44 as efficiently as the neutralizing anti-CD44 antibody used in the experiments for Fig. 7c–g (clone 515) (Supplementary Fig. 6J). Unlike mAb clone 515, mAb clone L178 did not inhibit and rather slightly enhanced virus capture by lnFRCs (Supplementary Fig. 6K), suggesting that the binding of a monoclonal antibody to CD44 itself is not likely to cause steric hindrance that reduces virus capture by lnFRCs.

Notably, we did not observe an additive effect on virus capture and trans-infection of A3.01 T cells when H-ase-treated virus was added to the anti-CD44 antibody-treated lnFRCs (Fig. 7e, f). Altogether, these results indicate that HA associated with virus particles and CD44 on the FRC surface are engaged in the same mechanism that promotes FRC-mediated virus capture and trans-infection, presumably through interactions with each other. These interactions likely contribute to not only virus capture but also virus retention on the FRC surface, since H-ase treatment of virus-loaded FRCs substantially reduced trans-infection (Supplementary Fig. 6L).

Finally, we found that lnFRCs capture HLAC-derived virus in a manner sensitive to H-ase treatment of virus, anti-CD44 antibody treatment of the virus or lnFRCs, and the addition of exogenous HA (Fig. 8a–d), indicating that FRCs capture viruses produced by T cells present in SLOs via the same molecular interactions promoting the capture of HeLa-derived viruses. Altogether, these results demonstrate that FRCs capture HIV-1 particles via the interactions between CD44 expressed on FRCs and HA bound to CD44 incorporated into virus particles, which likely leads to efficient virus spread in SLOs (Supplementary Fig. 7).

## Discussion
HIV-1 dissemination and production occur most efficiently in SLOs such as LNs[7,48,49]. Although most T cells in LNs reside in T cell zones where they constantly come into contact with FRCs, it has been unknown whether FRCs affect HIV-1 spread. In this study, we demonstrated for the first time that FRCs enhance HIV-1 spread among T cells via trans-infection. FRCs captured

both lab-adapted and primary transmitted/founder HIV-1 strains. The FRC-mediated trans-infection was observed with viruses produced by primary lymphocytes, likely CD4+ T cells, isolated from human tonsils. Furthermore, trans-infection was observed not only in 2D but also in 3D cultures of FRCs where FRCs form a 3D cell network resembling that in T cell zones. Therefore, it is likely that, in SLOs, FRCs capture virus particles released by infected T cells and then trans-infect uninfected T cells that come into contact, thereby contributing to efficient HIV-1 spread. While most in vivo studies on mechanisms promoting virus spread within SLOs have focused on interactions between hematopoietic cells[33,39,50,51], our findings highlight the need for investigating the roles played by lymphoid organ stromal cells in vivo.

Very recently, mucosal stromal fibroblasts were also shown to promote HIV-1 infection via trans-infection[52]. Therefore, fibroblastic cells are likely to play a previously unappreciated role in virus spread via trans-infection at SLOs as well as mucosal sites. However, it should be noted that in that study mucosal stromal fibroblasts were observed to capture 293T-derived viruses, which would not contain CD44 in their envelope membrane. Thus, FRC-mediated trans-infection in SLOs and trans-infection mediated by fibroblasts at mucosal sites likely differ in molecular mechanisms.

Memory T cells, including germinal center Tfh cells, which are likely to form major reservoirs for HIV-1 and SIV in vivo[31,32,53], express CD44 on their surface[54–56]. Therefore, it is likely that CD44 is incorporated into virus particles produced in vivo. Consistent with this possibility, anti-CD44 antibody interacts with HIV-1 virions isolated from the plasma of HIV-1 patients[57], and anti-CD44-bound microbeads can be used to promote sedimentation of viruses[58]. Although it has been long known that CD44 is incorporated into virus particles[34,59,60], the natural functions of HIV-1-incorporated CD44 in HIV-1 infection have not been clear. Here we have revealed a previously unrecognized function of such CD44 in virus spread; CD44 on virus particles interacts with CD44 on FRCs via virus-associated HA, which leads to virus capture and subsequent trans-infection mediated by FRCs.

Interestingly, HA present on the surface of CD44-expressing primary CD4+ T cells and T cell lines has been observed to

reduce direct binding and infection of CD44-containing HIV-1[46]. Steric hindrance of virus-receptor interactions by cell surface HA, which can have large molecular sizes, has been suggested as a potential mechanism[46,61]. However, considering that HA bound to virus particles binds to CD44 on the FRC surface (this study), an alternative or additional interpretation is possible for the effect of HA present on the T cell surface[46]; the T cell surface HA may have occupied CD44 on the surface, thereby blocking binding of virus particles to T cells via virus-associated HA. Consistent with this possibility, only a small fraction of primary T cells, either stimulated or non-stimulated by PHA, was observed to bind to exogenously added HA[62]. Thus, it is likely that whereas HA molecules bound to CD44 incorporated into virus particles promote binding to FRCs and thereby trans-infection, these virus-associated HA molecules would not participate in direct binding to target T cells unless T cell surface HA is removed or downregulated.

Since extracellular HA is internalized by endocytosis via its interaction with CD44[63], it is possible that HIV-1 particles captured by FRCs via HA are also internalized. This is of interest because studies of DC-mediated trans-infection suggest that HIV-1 particles need to be retained on the DC plasma membrane (or its invagination) but not endocytosed for the trans-infection[64–66]. We observed that FRCs pulsed with virus for 2 h and then treated with H-ase retain a fraction of the virus and mediate trans-infection of A3.01 T cells. This is consistent with a potential role in trans-infection for viruses not exposed on the cell surface during H-ase treatment.

It is conceivable that FRCs from different anatomical locations may express additional, CD44-independent mechanism for virus capture. Consistent with this possibility, the difference between capture of HeLa- and 293T-derived viruses was smaller with tFRCs than with lnFRCs (2.5–3 fold versus 5 fold; Figure 5a), and virus capture by tFRCs was less sensitive to anti-CD44 antibody treatment than that by lnFRCs (2–2.5 fold versus 5 fold; Figs. 7e, g). Likewise, we do not rule out the possibility that molecules other than HA and CD44 associated with virus particles promote capture by FRCs via an additional mechanism, since the host molecules incorporated into HIV-1 particles vary depending on producer cell types[34]. This may explain our observation that the capture of HLAC-derived virus by FRCs was less sensitive to perturbation of HA-CD44 interactions than the capture of HeLa-derived virus (2–3 fold versus 3–10 fold; Figure 6 and 7). Regardless of the presence of other mechanisms, however, our results obtained using viruses and FRCs from different sources demonstrate that the interaction between virus-associated HA and CD44 expressed on FRCs represent a common mechanism by which FRCs promote spread of HIV-1 produced by CD44-expressing cells.

Intriguingly, even though both FRCs and HeLa cells express CD44, we observed that trans-infection is mediated by FRCs but not by HeLa cells. CD44 is a transmembrane glycoprotein that are differentially glycosylated depending on the cell types[67]. In addition, many CD44 variant isoforms, which contain one or more variant exons, exist and are differentially expressed among cell types[68]. Furthermore, the differences in glycosylation and splicing patterns affect HA binding ability of CD44[69,70]. Therefore, it is quite possible that FRCs and HeLa express differentially glycosylated and/or spliced CD44 variants, which differ in virus capture efficiencies. Consistent with this possibility, we observed that the size of CD44 expressed by lnFRCs and HeLa cells are different (Supplementary Fig. 6H). It is also conceivable that a cofactor expressed in FRCs may be necessary for efficient virus capture. The molecular determinants that enable virus capture by CD44 expressed on FRCs will be a focus of future investigation.

Our study has identified the role of FRCs in HIV-1 spread and revealed a molecular mechanism by which FRCs capture HIV-1 and mediate trans-infection. Recent studies suggest that trans-infection by subcapsular sinus-lining macrophages or mucosal fibroblasts may promote acute phase infection[33,52]. Due to the location of FRCs in T cell zones and follicles, the FRC-mediated trans-infection is likely to contribute to HIV-1 dissemination in SLOs not only in the acute phase but also in subsequent periods, for example, upon interruption of antiretroviral therapy. In infected individuals, SLOs are likely to harbor latent viral reservoirs[11] and therefore may become early sites of productive infection in the event of latent virus reactivation[14]. Therefore, intervention of FRC-mediated trans-infection via the interactions between CD44 and HA could serve as a novel therapeutic strategy to suppress recurrence of HIV-1 spread within SLOs, especially since CD44-HA interactions are currently considered as a molecular target of an anti-cancer strategy[71]. However, such an antiviral strategy needs to be carefully evaluated in several model systems due to potentially diverse functions of CD44 or HA.

## Methods

**Cells and tissues**. A3.01 T cells (NIH AIDS Reagent Program, Cat# 166) were cultured in RPMI 1640 (Invitrogen) containing 10% fetal bovine serum (FBS; HyClone) and penicillin–streptomycin (P/S; Gibco) (RPMI-10). Human lymphatic fibroblasts (i.e., lnFRCs) (Sciencell Research Laboratories, Cat# 2530) were maintained in fibroblast medium (Sciencell Research Laboratories) containing 2% FBS (Sciencell Research Laboratories), P/S (Sciencell Research Laboratories), and fibroblast growth supplement (Sciencell Research Laboratories) (FM-2). These cells were used until passage 10. Immortalized lnFRCs (ilnFRCs) were generated by transfection of lnFRCs with pLXSN16E6E7 (Addgene), which encodes HPV E6 and E7, and subsequent selection in FM-2 containing 1000 μg/ml G418 (Gibco). A clone was isolated from G418-resistant cells by limiting dilution and cultured in FM-2 without G418. TZM-bl cells (NIH AIDS Reagent Program) were cultured in Dulbecco's modified Eagle's medium (DMEM) (Lonza) containing P/S and 10% FBS. HeLa cells (obtained from Dr. Eric Freed and verified by the AMPFLSTR Identifiler Plus Assay) and 293T cells (ATCC, Cat# 3216) were cultured in DMEM (Lonza) containing P/S and 5 and 10% FBS, respectively.

Tonsil tissue was removed from non-identifiable donors in routine tonsillectomy. The University of Michigan Medical School Institutional Review Board (IRB) has determined that the study does not fit the definition of human subjects research requiring IRB approval. Tonsil blocks were prepared, cultured, and infected according to a published protocol[72] with modifications. Briefly, tonsil tissue was dissected into approximately 8-mm³ blocks. The blocks were placed at the air–liquid interface of a Gelform (Pharmacia and Upjohn) soaked in 1 ml of tissue culture medium (RPMI-15) [RPMI 1640 containing 15% FBS, 1% fungizone (Gibco), 50 μg/ml gentamycin (Corning), 1% non-essential amino acids (Gibco), 1% sodium pyruvate (Gibco)] containing 0.31 μg/ml Timentin (United States Biological Corporation) in a 12-well plate. The blocks were cultured overnight, and the culture medium was replaced with RPMI-15 containing P/S prior to the inoculation.

HLACs were prepared, cultured, and inoculated with HIV-1 according to a published protocol[73] with modifications. Tonsil tissue was cut into approximately 8-mm³ blocks, grained, and passed through 40-μm cell strainers (Fisher Scientific). Living cells were then separated with ficoll from dead cells and resuspended in RPMI-15 containing P/S. The cells were plated in 96-well round bottom plates at a concentration of $2 \times 10^6$ cells/well and cultured for 1 or 2 days prior to the use in the experiments.

When indicated, tFRCs and HLACs were isolated from the same tonsils. Isolation of tFRCs was performed according to a published protocol[26] with modifications. In brief, tonsil tissue was cut into approximately 8-mm³ blocks and treated with 0.3 or 0.6 mg/ml collagenase P, 2.4 mg/ml Dispase II, and 0.3 mg/ml DNase I (Roche). The cell suspension, which contains stromal cells and lymphocytes, was collected and passed through 70-μm cell strainers (Fisher Scientific). Cells were resuspended in RPMI-15 containing P/S, plated, and cultured for 24–36 h. The adherent cells were washed with phosphate-buffered saline (PBS) twice to remove lymphocytes and cultured in RPMI-10 for 8 days. The cells were washed with PBS twice, detached from plates by treatment with PBS(−) containing 0.2% Trypsin, 5 mM EDTA for 2 min, and analyzed for the expression of CD45, CD31, and Podoplanin (PDPN) by flow cytometry. Approximately 85% of cells isolated in this protocol was FRCs based on the surface marker expression (CD45⁻, CD31⁻, and PDPN⁺).

Peripheral blood mononuclear cells (PBMCs) were obtained from buffy coats derived from healthy donors (New York Blood Center, NY), and PBLs were isolated from PBMCs by the plate depletion of CD14⁺ monocytes and the negative selection using CD14⁺ MicroBeads (Miltenyi Biotec) and cultured in RPMI-10 containing 6 μg/ml phytohemagglutinin (PHA; Sigma-Aldrich) to activate. To

obtain imDCs, CD14$^+$ monocytes were isolated from PBMCs by the positive selection using CD14$^+$ MicroBeads and cultured in RPMI-10 containing 100 ng/ml IL-4 (R&D systems) and 100 ng/ml granulocyte-macrophage colony-stimulating factor, (R&D systems) for 6 days. mDCs were differentiated from imDCs by culturing the latter in the presence of 100 ng/ml LPS (Sigma-Aldrich) for 48 h.

**ilnFRC encapsulation and 3D culture in PEG hydrogels**. Hydrogels were prepared with 8-arm PEG vinyl sulfone (PEG-VS, 40 kDa, >99% purity, Jenkem Technology) and formed via Michael-type addition with an MMP-sensitive bifunctional crosslinking peptide Ac-GCRD*VPMS↓MRGG*DRCG (VPMS) (1739 g/mol, >90% purity, GenScript, cleavage site indicated by ↓)[74]. PEG hydrogels were modified with the integrin binding peptide G*CGYGRGD*SPG (RGD) (1067.10 g/mol, GenScript) to allow attachment of the encapsulated cells.

Prior to in situ crosslinking and encapsulation, 4% PEG-VS solution was prepared by dissolving PEG in the appropriate amount of 0.05 M HEPES buffer at pH 7.4, and the monofunctional agents (0.75 or 1.5 mM RGD) were allowed to bind to the PEG macromers for 15 min. Meanwhile, ilnFRCs were suspended at a concentration of $2 \times 10^6$ cells per ml and centrifuged for 5 min at $300 \times g$ to remove excess media. The cell pellet was reconstituted with the modified PEG-VS solution, and the crosslinker (VPMS) was added to form a gel. The stoichiometric ratio of –VS to thiol (–SH) groups was kept constant at 1:1 ratio for all experiments. Each 60-μl gel was crosslinked in a 96-well plate at 37 °C for 30 min and flipped every 5 min to prevent cell sedimentation and clustering. After 30 min, gels were covered with 300 μl of FM-2. Media was changed every 3 days and cells were cultured up to 12 days.

**Plasmids**. A plasmid encoding standard isoform of human CD44 (CD44s), pCMV6-XL5/CD44s, was generated by standard molecular cloning techniques using pCMV6-XL5/CD44 (Origene), which encodes a full-length isoform (NM_001001391 [https://www.ncbi.nlm.nih.gov/nuccore/NM_001001391]). pLXSN16E6E7 (ref. [75]) was a gift from Denise Galloway (Addgene plasmid #52394). pSpCas9(BB)-2A-Puro[76] was a gift from Feng Zhang (Addgene plasmid #62988). For knocking out of CD44, a guide sequence of CD44 was synthesized and inserted into pSpCas9(BB)-2A-Puro [pSpCas9(BB)-2A-Puro/CD44-gRNA] according to the Zhang laboratory protocol[76]. The guide sequence was 5′-GTCGCTACAGCATCTCTCGGA-3′.

**Virus stocks and HIV-1 infection**. Virus stocks of the lab-adapted strain NL4-3 and its Env-deficient derivative NL4-3/KFS, primary transmitted/founder (T/F) strains WITO, CH040, CH077, CH086, CH106, RHPA, THRO, TRJO, and SUMA[77,78] and a chimeric infectious HIV-1 subtype C molecular clone MJ4 (ref. [35]) were prepared by transfecting HeLa or 293T cells, two cell lines commonly used as virus producer cells, with corresponding molecular clones using Lipofectamine2000 (Invitrogen). To generate HIV-1$_{NL4-3}$ in HeLa cells lacking CD44 expression, HeLa cells were transfected with pSpCas9(BB)-2A-Puro/CD44-gRNA and cultured for 24 h. Subsequently, the cells were cultured for 3 days in the presence of 2 μg/ml puromycin (Sigma-Aldrich), and puromycin-selected cells, used as CD44-knocked out cells, were transfected with pNL4-3. To generate HIV-1$_{NL4-3}$ in CD44-reconstituted HeLa cells, CD44-knocked out HeLa cells were co-transfected with pCMV6-XL5/CD44s and pNL4-3. To generate HIV-1$_{NL4-3}$ in CD44-overexpressing 293T cells, 293T cells were co-transfected with pCMV6-XL5/CD44s and pNL4-3. Two days post-transfection, virus-containing supernatants were collected. To generate HLAC-derived virus, HLACs were inoculated with 50 ng p24 of 293T-derived HIV-1$_{NL4-3}$ by spinoculation at 2500 rpm (888 or $1462 \times g$ depending on the centrifuge used) for 2 h, placed at 37 °C, and cultured for 2–16 h. The infected HLACs were washed by RPMI-10 three times, and the culture medium was collected on day 5 or 6. These virus-containing supernatants were filtered through a 0.45-μm filter and used as virus stocks. For quantification of virus, the p24 amount was measured by ELISA. For some experiments, virus-containing supernatant was concentrated by ultracentrifugation (24,000 rpm for 2 h at 4 °C) using a 20% Sucrose/PBS cushion.

Tonsil blocks were inoculated with 25.4 ng p24 virus by depositing 5 μl virus stocks on top of each block[72].

**p24 ELISA**. p24 ELISA was performed as described previously[79]. Briefly, virus-pulsed cells and supernatant containing viral particles were lysed in ELISA lysis buffer (0.05% Tween 20, 0.5% Triton X-100, 0.5% casein in PBS). Anti-HIV-1CAp24 antibody (1:1000 or 1:2000; clone 183-H12-5C; NIH AIDS Research and Reference Reagent Program, Cat# 3537) was bound to Nunc MaxiSorp plates overnight. Lysed samples were captured for 2 h and incubated with biotinylated antibody to HIV-1CAp24 (1:2000 or 1:4000; clone 31-90-25; ATCC, Cat# HB-9725). 31-90-25 was biotinylated with the EZ-Link Micro Sulfo-NHS-Biotinylation Kit (Pierce). Samples were detected using streptavidin-HRP (Fitzgerald) and 3,3′,5,5′-tetramethylbenzidine substrate (Sigma-Aldrich). CAp24 concentrations were determined using recombinant HIV-1 IIIB p24 recombinant protein for standards (NIH AIDS Research and Reference Reagent Program).

**Analysis of infectivity**. To analyze infectivity of HeLa- and 293T-derived viruses, $1 \times 10^4$ TZM-bl cells were inoculated with 76.1 ng p24 viruses and cultured in the presence of 10 μM Saquinavir (NIH AIDS Research and Reference Reagent Program), which prevents multi-round replication. After 48 h post-infection, cells were lysed, and the cell lysates were analyzed for luciferase activity using a commercially available kit (Promega).

**Immunoblotting**. For analysis of CD44 incorporation into virions, HeLa, CD44-knocked out HeLa, and 293T cells were transfected as described above (see Virus stocks and HIV-1 infection). These transfected cells were cultured for 2 days. Cell and virus lysates were prepared as previously described[80]. These lysates were subjected to sodium dodecyl sulfate polyacrylamide gel electrophoresis, followed by immunoblotting using anti-CD44 (1:500; clone 2C5; R&D systems, Cat# BBA10), anti-HIV Ig (1:1000; NIH AIDS Research and Reference Reagent Program, Cat# 3957), or anti-tubulin (1:4000; clone B-5-1-2; Sigma-Aldrich, Cat# T5168) followed by corresponding secondary antibodies conjugated to horseradish peroxidase (1:10,000; Santa Cruz Biotechnology, Cat# sc-2005 and sc-2453; or 1:100,000; EMD Millipore, Cat# AP130P). Signals were detected using SuperSignal West Pico chemiluminescent substrate (Pierce) or SuperSignal West Femto chemiluminescent substrate (Pierce).

**Flow cytometry**. Cells were fixed by 4% PFA/PBS for 30 min at 4 °C. The fixed cells were immunostained using following primary antibodies: anti-CD43 (1:100; clone 1G10; BD Biosciences, Cat# 555474), anti-CD44 (1:100 for clones 515 and 1:200 for L178; BD Biosciences, Cat# 550988 and 559046, respectively), anti-PSGL-1 (1:100; clone KPL-1; BD Biosciences, Cat# 556053), or anti-CD169 (1:100; clone 7D2; Abcam, Cat# ab18619). Alexa Fluor 488- and Alexa Fluor 647-conjugated anti-Mouse IgG (H + L) were used as secondary antibodies (1:200; Invitrogen, Cat# A-11001 and A-21235, respectively). The following antibodies primarily labeled with a fluorophore were also used: anti-podoplanin-PE (1:100; clone NZ-1; AngioBio, Cat# 11-009PE), anti-CD45-APC (1:100; clone HI30; BioLegend, Cat# 304011), anti-CD31-FITC (1:100; clone WM59; BioLegend, Cat# 303103), anti-CD3-APC (1:100; clone UCHT1; BD Biosciences, Cat# 555335), anti-CD45RO-PE (1:100; clone UCHL1; BioLegend, Cat# 304206), anti-DC-SIGN-FITC (1:100; clone DCN46; BD Biosciences, Cat# 551264), anti-CD11c-Alexa647 (1:100; clone 3.9; BioLegend, Cat# 301619), and anti-CD83-PE (1:100; clone HB15e; BioLegend, Cat# 305307). To detect IL-7 in lnFRCs, adherent cells were detached from plates by treatment with Dispase II (Roche) in DMEM containing 5% FBS or 0.2% Trypsin containing 5 mM EDTA, fixed with 4% PFA/PBS, and permeabilized with 0.1% Triton X-100 for 5 min at 4 °C. The cells were immunostained with anti-IL-7 (1:50; clone 7417; R&D systems, Cat# MAB207) followed by Alexa Fluor 488-conjugated anti-Mouse IgG (H + L) (1:200; Invitrogen, Cat# A-11001). To detect intracellular Gag in infected cells, fixed cells were permeabilized by 0.1% Triton X-100 for 5 min at 4 °C and stained using anti-HIV Gag-FITC (1:200; clone KC57: Beckman Coulter, Cat# 6604665).

**Detection of cell surface HA**. lnFRCs were either untreated or treated with 1 mM 4-methylumbelliferone (Sigma-Aldrich) for 1 day at 37 °C, fixed with 4% PFA/PBS for 30 min and treated with biotinylated hyaluronic acid binding protein (1:100; Calbiochem, Cat# 385911) overnight at 4 °C. The cells were subsequently treated with Alexa594-streptavidin (1:200; Invitrogen, Cat# S11227) for 1 h at room temperature and washed by PBS twice. The fluorescence signal was detected using a Nikon TE-2000U inverted epifluorescence microscope (Nikon).

**Comparison of ilnFRCs morphologies in 2D and 3D conditions**. For the 2D condition, ilnFRCs were plated, cultured for 7 days, and observed using the Nikon TE-2000U inverted epifluorescence microscope. For the 3D condition, ilnFRCs were encapsulated into PEG gels as described above and cultured for 14 days. The cells were observed using a Leica DMI3000 B inverted microscope (Leica).

**Confocal microscopy analysis**. ilnFRCs encapsulated into PEG gels and cultured for 14 days were stained with 25 μM CellTracker CMFDA (Invitrogen) by direct addition of the dye to the gels. Following incubation for 45 min at 37 °C, the gels containing encapsulated ilnFRCs were washed with PBS twice. In parallel, A3.01T cells were stained by 12.5 μM CellTracker CMTPX (Invitrogen) for 30 min at 37 °C and washed with PBS twice. The stained A3.01T cells were added to the PEG gels encapsulating stained ilnFRCs and cocultured for 3 days. The cells were visualized using a Nikon A1 confocal microscope with a ×10 objective lens (Nikon). Z-series of images were acquired with 5.92 μm intervals between focal planes. A maximum intensity projection image and a 3D reconstructed image of the z-series images composed of 40 focal planes were obtained with ImageJ software (NIH; downloaded from http://rsbweb.nih.gov/ij/).

**Analysis of HIV-1 replication kinetics**. $4 \times 10^5$ A3.01 T cells were inoculated with 0.254 ng p24 of HIV-1$_{NL4-3}$ in the presence or absence of lnFRCs in 1 ml RPMI-10. To analyze infection of lnFRCs, lnFRCs were also inoculated in the absence of A3.01 T cells with 0.254 ng p24 of HIV-1$_{NL4-3}$. The 50-μl culture medium was subsequently collected every 2 days, and mixed with ELISA lysis buffer. p24 amounts in the lysates were determined by p24 ELISA. Remaining medium and A3.01 cells were gently resuspended, and the 700 μl of the cell suspension was

discarded. To maintain the cultures, 750 µl of fresh RPMI-10 was added to each wells. During the experimental period, lnFRCs were not detached but kept in the culture.

**Cell-to-cell transfer/transmission assay.** $2 \times 10^5$ A3.01 T cells were inoculated with HIV-1$_{NL4-3}$ (approximately 1.2 µg p24). Five days post-infection, the cells were stained with 3 µM CellTracker CMTMR (Invitrogen) for 15 min at 37 °C, washed with PBS, and further cultured for 1 day. The stained cells were washed with PBS and used as donor cells. $6 \times 10^5$ A3.01 T cells and $3 \times 10^5$ CMTMR-stained donor cells were either cocultured or cultured separately using transwell in the presence or absence of lnFRCs. After 2 days, T cells were fixed with 4% PFA/PBS, permeabilized with 0.1% Triton X-100, and immunostained with anti-HIV Gag-FITC (clone KC57; Beckman Coulter). The infected cells were analyzed by flow cytometry. T cells were distinguished from lnFRCs by forward and side scatter, as T cells were much smaller and less granular than lnFRCs.

**Virus capture assay.** $5 \times 10^4$ lnFRCs, tFRCs, ilnFRCs, and HeLa cells were plated in 24-well plates. Sixteen to 24 h after plating, at which cells are approximately 80% confluent, the cells were pulsed with HIV-1$_{NL4-3}$ (7.6, 25.4, or 100 ng p24) for 2 h. Virus stocks contain either 5, 10, or 15% FBS; however, we have confirmed that virus capture was not affected by the presence and absence of 10% FBS during incubation of cells with virus. After removal of unbound virus through washing with PBS four times, cells were lysed in the ELISA lysis buffer, and the lysates were examined for the amount of captured virus in p24 ELISA assays. To test the effect of blocking of cell surface CD44 on virus capture, cells were washed with PBS, pre-incubated with 10 µg/ml anti-CD44 antibody (clone 515 or L178; BD Bioscience) or mouse IgG1 isotype control (BD Biosciences) for 1 h at 4 or 37 °C, and then washed with PBS twice prior to addition of virus. For testing the role of CD44 incorporated into virus particles, HIV-1$_{NL4-3}$ was pre-incubated with 0.625 µg/ml anti-CD44 antibody (clone 515; BD Bioscience) or mouse IgG1 isotype control (BD Biosciences) for 1 h at 37 °C and used without removing excess antibody. To examine the effect of removal of HA from the surface of lnFRCs, lnFRCs were pretreated with 10 units/ml H-ase (Sigma-Aldrich) in the culture medium for 1 h at 37 °C and washed with PBS twice prior to addition of virus. To test the effect of removal of HA from virus surface, virus were pre-incubated with 10 units/ml (for 7.6 and 25.4 ng p24 of virus) or 25 units/ml (for 100 ng p24 of virus) H-ase for 1 h at 37 °C and used without separating virus from H-ase. To investigate the effect of exogenous HA on virus capture, cells were pulsed with virus in the presence of 2.5 mg/ml HA (molecular weight <10 kDa and 100–150 kDa: Lifecore Biomedical).

In 3D culture, ilnFRCs encapsulated in PEG gels were pulsed with 7.6 ng p24 HIV-1$_{NL4-3}$ for 16 h and washed with RPMI-10 four times. The ilnFRCs in the gels were further cultured for 4 h and washed with RPMI-10 twice. The gels were digested with liberase (Sigma-Aldrich), and the digested gels were mixed with ELISA lysis buffer prior to measurement of captured p24 amounts in p24 ELISA assays.

**Trans-infection assay in 2D and 3D culture systems.** lnFRCs, tFRCs, ilnFRCs, imDCs, mDCs, and HeLa cells were plated and pulsed with 7.6 or 25.4 ng p24 NL4-3 for 2 h as described in the Virus capture assay section. After removal of unbound virus through washing with PBS or RPMI-10, cells were overlaid with $4 \times 10^5$ A3.01 T cell line or $1.5 \times 10^6$ PHA-stimulated PBLs. A3.01T cells and PBLs were cocultured with virus-pulsed cells for 6 and 4 days, respectively. A3.01T cells and the PBLs were harvested, fixed with 4% PFA/PBS, and permeabilized with 0.1% Triton X-100 in PBS. Infected cell numbers were determined using anti-HIV Gag-FITC (clone KC57; Beckman Coulter) in flow cytometry. To compare trans-infection efficiency of lnFRCs, imDCs, and mDCs, harvested cell suspensions were also immunostained with anti-CD3 antibody so as to distinguish A3.01T cells from DCs. To distinguish memory T cells from other cells among PBLs, anti-CD3 and anti-CD45RO antibodies (clones UCHT1; BD Biosciences and UCHL1; BioLegend, respectively) were used. Treatment with anti-CD44 antibody, H-ase, or exogenous HA was performed as described in the Virus capture assay section.

To examine the extent of trans-infection in the 3D culture, encapsulated ilnFRCs were pulsed with 7.6 ng p24 NL4-3 for 16 h and washed with RPMI-10 four times. The ilnFRCs were further cultured for 4 h and washed with RPMI-10 twice. $1 \times 10^5$ A3.01T cell line was added to the gels and cocultured for 6 days. After 3 days of coculture, a half of culture medium was replaced with fresh medium. The p24 values in the supernatant were determined by p24 ELISA.

**Statistical analyses.** Statistical analyses were performed by Prism 7.0 (Graphpad software). For Fig. 2a, the data meet the assumption of sphericity. For two-group comparisons, either two-tailed paired Student's $t$-test and Wilcoxon matched-pairs signed rank test was used. For multi-group comparisons, Tukey's test following one-way ANOVA was used. The applied statistical analyses are reported in the figure legends. Data are presented as mean ± SD or SEM. and were considered statistically significant when the $p$-value was <0.05.

**Data availability.** The data that support the findings of this study are available from the corresponding author upon request.

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

## Acknowledgements

We thank Drs. Kathleen Collins and Eric Freed and members of our laboratories for helpful discussions and critical reviews of the manuscript. We thank Dr. Collins for non-biotinylated and biotinylated anti-HIV-1CAp24 (clone 31-90-25). We also thank Dr. Freed for HeLa cells. The following reagents were obtained through the AIDS Research and Reference Reagent Program, Division of AIDS, NIAID, NIH: Saquinavir, the panel of full-length transmitted/founder HIV-1 infectious molecular clones from Dr. John Kappes, HIV Ig from NABI and NHLBI, HIV-1 IIIB p24 recombinant protein, and HIV-1 p24 monoclonal antibody (183-H12-5C) from Drs. Bruce Chesebro and Kathy Wehrly. We thank Drs. Feng Zhang and Denise Galloway for pSpCas9(BB)-2A-Puro and pLXSN16E6E7, respectively, both of which were obtained through Addgene. This work is supported by the University of Michigan MCubed seed fund, NIH grants R01 AI071727 (to A.O.), R21 AI120815 (to A.O.), and T32 DE007057 (to J.K.).

## Author contributions

T.M., J.K., and Y.L. performed most experiments and analyzed the data. T.M., J.K., Y.L., A.S., and A.O. conceived and designed the experiments. G.E.G. provided materials essential for the study. T.M. and A.O. wrote the manuscript.

## Additional information

**Competing interests:** The authors declare no competing interests.

