## [Peer Review File · Nature Communications]

Reviewers' comments:

Reviewer #1 (Remarks to the Author):

The manuscript by Murakami et al explores the hypothesis that lymph node stromal cells such as fibroblastic reticular cells (FRCs) can mediate efficient HIV-1 trans infection of CD4+ T cells that is dependent on an unusual virus capture mechanism; hyaluronan dependent bridging of CD44 receptors expressed on FRCs and CD44 incorporated within virus particles. Abrogation of CD44 incorporation in virus particles, treatment of virus particles with hyaluronidase or neutralizing CD44 on FRCs upon pretreatment with monoclonal antibodies all prevented HIV-1 capture by FRCs and FRC-dependent trans infection. Additionally, close cell-to-cell interactions, but not soluble factors secreted by FRCs, was necessary for FRC-mediated enhancement of T cell infection. The authors observe a producer cell dependence on virus capture by FRCs, since HIV-1 derived from transfected HeLa (CD44 positive) cells, or virus propagated in human lymphoid tissue aggregates (HLACs) but not from HEK293T cells (CD44 negative) was captured efficiently by FRCs in a CD44-dependent manner. These are interesting findings and provide a mechanism by which lymph node resident stromal cells can facilitate infection of lymphoid-tissue resident CD4+ T cells.

The strengths of the manuscript are the use of primary FRCs and HLFs in cell-to-cell co-culture assays and use of immortalized HLFs in 3-D cultures to mimic the in vivo SLO environment, and the use of HLAC-passaged virus for some of the studies. But to the detriment of the manuscript, the authors primarily utilize A3.01 CD4+ T cell line for all of their studies, which are not at all representative of the CD4+ T cell subsets in secondary lymphoid tissues. In the only experiment utilizing primary T cells for trans infection studies (Fig. 4F), the extent of FRC-mediated enhancement was at best 2-fold, questioning the significance of FRC-mediated HIV trans infection in vivo. The conclusions would be greatly strengthened by the use of primary CD4+ memory T cells for all of the trans infection assays (the cell type that is the predominantly infected in vivo), and determining if the robust enhancements observed with A3.01 T cell lines are also recapitulated with primary T cells.

Specific points for the authors to consider:

1. The authors need to knock-down CD44 expression in primary HLFs or FRCs. While data is shown with CD44 knock-out HeLa cells, similar strategies are needed with primary HLFs or FRCs to conclusively prove the role of FRC-associated CD44 in virus capture. The use of monoclonal antibodies to neutralize CD44 in FRCs is problematic because of non-specific steric hindrance that might reduce virus capture and transfer independently of CD44 function.

- a. Additionally, the authors need to demonstrate CD44 expression on FRC surface. The authors only show data with HLFs or cell lines but not with FRCs. This is central to their hypothesis.

2. While HLFs display a robust 5-fold difference in capture efficiency between HEK293T and HeLa cell derived viruses (Fig. 4), and blocking CD44 on HLFs results in ~5-fold reduction in capture and transfer, these findings are not recapitulated with FRCs (cell type most relevant

to the argument put forward by the authors), where there is only 2-2.5 fold difference in capture between HeLa and HEK293T derived virus particles (Supplementary figure 3A). Additionally, pre-treatment of FRCs with anti-CD44 antibodies only resulted in a ~2-fold decrease in virus capture. In contrast to HeLa-derived virus, capture of HLAC-derived virus was only modestly inhibited by HA treatment, or blocking CD44 function on HLFs (Supplementary figure 4). These findings should be included in the main text of the manuscript and the caveats of these findings discussed.

3. Note that CD44 is expressed ubiquitously in most cells in vivo. HeLa cells express CD44 at levels similar to that of HLFs, based on FACS profiles presented in Fig 5 and supplementary figure 4. But the capture efficiency of HIV-1 particles by HeLa cells is significantly lower than that observed with HLFs (Fig. 1E). And, HeLa cells fail to transfer HIV to T cells. What is the basis for the cell-type specificity for CD44-mediated capture and transfer of HIV? Do HLFs (or FRCs) express additional cell surface molecules that are necessary for the virus capture and transfer mechanism?

4. Similar to the capture of HIV-1 by Siglec1 (mediated by incorporation of host glycosphingolipids), HLF-mediated virus capture was dependent on incorporation of HA-bound host CD44 molecules. This is an Env-independent virus capture mechanism, but surprisingly some virus isolates (TRJO and SUMA) were captured with higher efficiency than NL4-3 (Fig 4B). What is the reason for this difference?

5. HA is expressed as both low molecular weight (LMW) and high molecular weight (HMW) forms with differing abilities to cluster cell surface receptors, such as CD44, with differing consequences on downstream signaling. Does the type (LMW or HMW) of HA conjugated to CD44 on virus particles impact FRC/HLF mediated trans infection?

6. Previous findings on cell-associated HIV transfer, mediated by DC-SIGN, Siglec1 or mucosal stromal fibroblasts (unknown receptor), have suggested requirement for retention of captured virus at the cell surface. What is the mechanism of HLF (or FRC) mediated HIV trans infection? Does captured virus remain externalized on the HLF (FRC) surface? CD44 is a type I transmembrane protein which can internalize extracellular HA. Does virus captured by CD44 expressed on HLFs (FRCs) endocytosed? If not, does virus particle retention at the cell surface critical for HLF (FRC)-mediated trans infection?

7. The authors utilize FRCs and HLACs from multiple donor tonsillectomies for their studies (Fig 2). But whether HLFs were derived from multiple independent donors is not clear. The authors should provide information on whether HLFs (bought from a commercial vendor) was repeated with multiple independent stocks of HLFs.

Reviewer #2 (Remarks to the Author):

The manuscript by Murakami and colleagues describes the impact of CD44-HA interaction

incorporated in viral particles with cell surface CD44 of FRCs, and the subsequent enhanced CD44-dependent trans-infection of CD4+ T cells by these FRCs. While trans-infection by mucosal stromal fibroblasts was recently found, the current results are important and novel, because not only the mechanism for trans-infection (different from the mucosal fibroblasts) is now completely explored and explained, also the fibroblastic cell source is different, being FRCs from secondary lymphoid organs.

The strength of the manuscript is that with many different techniques, ie. CD44 blocking antibodies, H-ase, CD44 overexpression and CRISPR/Cas9-mediated deletion of CD44, and with different sources of primary human FRCs, the role for CD44-HA interaction and subsequent CD44-mediated virus capture and trans-infection has been very systematically demonstrated. In addition, the mechanism was also proven to work for at least 9 primary/founder HIV-1 clones. As such, the findings are important and demonstrate a new mechanism through which HIV-1 could spread in vivo in humans, clearly influencing current therapies.

Notably, however, to elevate the message even more, it would be nice to demonstrate the source (real) of CD44-HA virus particles in vivo (or ex vivo), which was suggested to be CD4+ T cells in the HLAC fraction from tonsils (line 261), but HA could also come from FRCs (Supplemental Figure 4D). In addition, while A3.01 seems to be a nice model for CD4+ T cells, the results would greatly benefit from the use of primary CD4+ T cells instead, as A3.01 is a T cell acute lymphoblastic leukemia. The choice for A3.01 should at least be discussed, and some experiments (eg. experiments such as Figure 1C-G) could be performed with primary human CD4+ T cells as well (see Neidleman et al, 2017). Lastly, while the mechanism is very nicely untangled, it would be interesting to demonstrate the trans-infection efficiency of FRCs as compared to (or together with) DCs. This would strengthen the important in vivo role of FRCs (compared to DCs) in secondary lymphoid organs in humans in the spread of HIV-1 even more. This latter point is interesting as DCs and FRCs, both interacting with T cells, also closely contact each other in secondary lymphoid organs.

How does this all fit in the model proposed in Supplementary Figure 5?

Major comment:

What is the source of HA? Although the results show that HA CD44 is produced by viral particle producing cells (mostly HeLa, Figure 5), there is only indirect/circumstantial evidence that CD4+ T cells (A3.01) can produce it themselves. As such, the authors state that it is likely CD4+ T cells isolated from tonsils (based on Figure 2). This is unfortunate, as it is important for a complete model to identify the HA producers (primary CD4+ T cells, FRCs, or both). Figure 5H implies that HA on FRCs inhibits capture efficiency, as it is increased upon H-ase treatment of HLFs. Yet, it seems counterintuitive (and weakening the message) that this is really the case in vivo. It seems more logical that CD4+ T cells produce viral particles with CD44 (either or not) lacking HA, and that FRCs capture these particles with HA-CD44 on their surface to subsequently infect uninfected T cells. To better understanding, the method of H-ase treatment should be described more extensively.

Details of washing steps are missing, and also the difference between treatment of cells versus virus. Also, when was 25 units/mL Hase used (line 516)? Could experiments in Figure 1C/D be combined with Hase treatment (and primary CD4+ T cells)?

Minor comments:

1. Figure 1B: it is somewhat unclear from the legend that analysis for p24 was performed on culture medium. This is always in the presence of p24 of HIV-1(NL4-3), and either or not in presence of A3.01 T cells. Legend should more clearly state what is shown in the graph without deducting it from the experimental procedure. Also, are the adherent cells detached for p24 analysis (and replaced afterwards)? How much culture medium was collected and what did it contain (cells or medium only, or both)?

2. Figure 2A-B: Virus obtained from HLAC was used to study trans-infection by HeLa/HLFs/FRCs, but what are the target cells? Primary CD4 or A3.01 T cells? This is not clear from any resource. Why is "donor-matched" HLACs important (or not)?

3. Line 140: what is the evidence in Figure 1 and 2 that demonstrate that FRCs mediate trans-infection of HIV-1 produced by T cells present in SLOs?

4. Figure 4/Supplemental Figure 3: why was a P2 clone of A3.01 T cells used instead of parental? What is the difference of P2?

5. Figure 4F: what type of T cells were infected and measured? Is also unclear from procedure.

6. Supplemental Figure 3C, D and H: significant differences?

7. Figure 5D, G and J: what are the target cells for trans-infection?

8. Many details such as time of inoculation, co culture etc. are mentioned in the experimental procedures (and Figure 1 only), however, some details should also be mentioned in the legends for clarity and convenience of the reader without going back-and-forth to the methods section.

Suggestion for the title: start with "Secondary..."

Line 139: add "T" to "CD4+ cells"

Line 178: T cells zone = T cell zone

Line 218: play = plays

Line 224: particle = particles

Line 235: a HA = an HA

Line 825: Date = Data

Line 834: in C. add "of A3.01 T cells" (this is the case throughout manuscript as mentioned before)

Reviewer #3 (Remarks to the Author):

Murakami and colleagues present data demonstrating that a stromal cell found in secondary lymphoid organs (SLO), the fibroblastic reticular cell (FRC) is capable of enhancing HIV infection of Tcells through a "trans-infection" mechanism that depends on the binding of virions through interactions between the ECM receptor CD44 present on virions and on the surface of both FRC and target Tcells using hyaluronan as a bridging ligand. These facilitating interactions may be particularly important since SLO are key sites for HIV dissemination and production. The magnitude of the enhancement effects are at first seemingly modest (2 to 6 fold) but it is of interest that the % infected cells in some of the models presented are quite impressive with up to 60% of target cells infected. The paper is well written and the data are carefully presented and shed light on a novel cell-cell interaction that could have particular relevance for viral dissemination and propagation in vivo. These results and the resulting model should be of interest to the journal readership. However, some additional information would increase the value of these observations to the field.

The experimental models are based on use of the lymphocytic cell line, A3.01 and HLAC (human lymphoid aggregate cells) derived from healthy donor tonsillar tissue as target cells for HIV. There is value in the use of HLAC cells since they reflect a primary cell population that seems like it would be relevant to this trans-infection system. At the same time, it would be valuable to know whether primary CD4+-Tcells such as those in peripheral blood would yield similar results so as to place these results in the context of the large experience and voluminous observations with in vitro models of primary CD4+Tcell infection. Related to this, the HLAC cells would presumably be a mix of many different cell types including CD8 Tcells, B cells etc. Have the authors found it necessary that all these different cell types be present in order for to see the enhancement of effect of CD44 expressing FRC? Would isolated CD4+TCells from the HLAC mix show the same effect?

In addition to the laboratory adapted NL4-3 and env- NL43 derivative, the authors studied a panel of primary, "transmitter-founder" viral clones and found similar enhancement effects for trans-infection. These transmitter-founder clones may have unique characteristics not found in primary viral isolates from those with established chronic infection. Did the authors have an opportunity to test other, "non-transmitter-founder" viral clones for comparison? Was the exclusive use of the transmitter founder panel because the trans-infection enhancement was unique to these viruses?

In Figure Supp 4H, exogenous HA is shown to reduce infection efficiency. The HA treatment conditions are not described in the figure legend or methods and should be. Additionally, Fetal calf serum contains considerable amounts of HA and appears to have been an additive to all (or nearly all) the culture media. Can the authors address how this might have influenced experimental results in this and other experiments? Both the HA sizes added in this experiment are relatively small (<10kd and 100 to 150kd). Did the authors have an opportunity to determine whether High molecular weight HA would have a comparable inhibitory effect?

Point-by-point Response to Referees

We thank the reviewers for the helpful and constructive comments. In response, in the revised manuscript, we included several new data obtained through new experiments as new figures (Figures 2A, 2B, and 5A and Supplementary Figures 2A, 2B, 4A, 4F, 5F, 5I, and 5J), described more details on experimental procedures, and included new points in the Discussion section, all of which we think improved the manuscript substantially. Our responses to each of the reviewers' comments are described below under the headings of **[Response]**.

Reviewer #1 (Remarks to the Author):

The manuscript by Murakami et al explores the hypothesis that lymph node stromal cells such as fibroblastic reticular cells (FRCs) can mediate efficient HIV-1 trans infection of CD4⁺ T cells that is dependent on an unusual virus capture mechanism; hyaluronan dependent bridging of CD44 receptors expressed on FRCs and CD44 incorporated within virus particles. Abrogation of CD44 incorporation in virus particles, treatment of virus particles with hyaluronidase or neutralizing CD44 on FRCs upon pretreatment with monoclonal antibodies all prevented HIV-1 capture by FRCs and FRC-dependent trans infection. Additionally, close cell-to-cell interactions, but not soluble factors secreted by FRCs, was necessary for FRC-mediated enhancement of T cell infection. The authors observe a producer cell dependence on virus capture by FRCs, since HIV-1 derived from transfected HeLa (CD44 positive) cells, or virus propagated in human lymphoid tissue aggregates (HLACs) but not from HEK293T cells (CD44 negative) was captured efficiently by FRCs in a CD44-dependent manner. These are interesting findings and provide a mechanism by which lymph node resident stromal cells can facilitate infection of lymphoid-tissue resident CD4⁺ T cells.

The strengths of the manuscript are the use of primary FRCs and HLFs in cell-to-cell co-culture assays and use of immortalized HLFs in 3-D cultures to mimic the in vivo SLO environment, and the use of HLAC-passaged virus for some of the studies. But to the detriment of the manuscript, the authors primarily utilize A3.01 CD4⁺ T cell line for all of their studies, which are not at all representative of the CD4⁺ T cell subsets in secondary lymphoid tissues. In the only experiment utilizing primary T cells for trans infection studies (Fig. 4F), the extent of FRC-mediated enhancement was at best 2-fold, questioning the significance of FRC-mediated HIV trans infection in vivo. The conclusions would be greatly strengthened by the use of primary CD4⁺ memory T cells for all of the trans infection assays (the cell type that is the predominantly infected in vivo), and determining if the robust enhancements observed with A3.01 T cell lines are also recapitulated with primary T cells.

[Response] We thank the reviewer for the helpful comments. To address the reviewer's comment on target cells, we analyzed FRC-mediated trans-infection using PHA-stimulated peripheral blood leukocytes (PBLs) as target cells. We observed that HLFs mediate trans-infection of memory T cells positive for CD45RO. The enhancement of

HIV-1 infection based on the increase in % Gag-positive cells at 4 days post-infection was about 2-fold relative to cell-free infection. It is worth noting that in this comparison, the amount of virus present in the 4-day cultures for cell-free infection was 100 ng (p24), since we did not remove the inoculum. In contrast, typically ~2 ng of virus (i.e., virus captured by HLFs) remains after 2-h inoculation and subsequent washing on day 1 in the trans-infection culture. Therefore, that we observed the 2-fold enhancement of PBL infection in the trans-infection cultures relative to cell-free infection cultures despite that ~50 fold less virus was present underlines the high efficiency of trans-infection mediated by HLFs (FRCs). The outcomes of the PBL coculture experiments (shown as new Figure 2A and B) and the interpretation are described in the Result section (lines 138-161). We also included the data for the amount of virus captured by HLFs when 100 ng (p24) virus was used as the inoculum (shown as new Figure 5A center panel). Because there was a donor-to-donor difference in the extent of HLF-mediated enhancement of infection among PBLs, we chose to use A3.01 cells rather than PBLs for subsequent experiments aimed at verifying that viruses captured by FRCs are transmittable. This point is also noted in the Result section (lines 157-161).

Specific points for the authors to consider:

1. The authors need to knock-down CD44 expression in primary HLFs or FRCs. While data is shown with CD44 knock-out HeLa cells, similar strategies are needed with primary HLFs or FRCs to conclusively prove the role of FRC-associated CD44 in virus capture. The use of monoclonal antibodies to neutralize CD44 in FRCs is problematic because of non-specific steric hindrance that might reduce virus capture and transfer independently of CD44 function.

[Response] We attempted to introduce the CRISPR/Cas9 system targeting CD44 into HLFs via transfection, nucleofection and transduction. However, we could not obtain CD44-negative HLFs. Since CD44 is involved in proliferation of human fibroblasts [1, 2], it is possible that CD44 knock-out HLFs may have failed to grow. Here the reviewer is concerned about the possibility that binding of the anti-CD44 monoclonal antibody to FRCs may inhibit virus capture not because the antibody blocks the CD44 function, i.e., binding to HA, but because the antibody causes steric hindrance. To address this concern by an alternative approach, we tested whether another monoclonal antibody that binds CD44 via a region unrelated to HA binding inhibits capture of HeLa-derived virus by HLFs. We used an anti-CD44 monoclonal antibody (clone L178), which binds to CD44 as efficiently as the neutralizing anti-CD44 antibody (clone 515) used in the original experiments but recognizes a different epitope in an extracellular domain of CD44. Using the virus capture assay, we observed that the clone 515 inhibited virus capture by HLFs as shown in Fig. 6J and K, but the clone L178 failed to inhibit virus capture despite that it bound slightly more abundantly to the HLF surface than the clone 515. Therefore, the binding of monoclonal antibody to CD44 is not likely to reduce virus capture by HLFs via steric hindrance. We added the data as new Supplementary Fig. 5I and J and described the result in the Results section (lines 290-300).

a. Additionally, the authors need to demonstrate CD44 expression on FRC surface. The authors only show data with HLFs or cell lines but not with FRCs. This is central to their hypothesis.

[Response] We confirmed CD44 expression on the surface of FRC isolated from tonsils of 5 different donors. We added the data as new Supplementary Fig. 5F.

2. While HLFs display a robust 5-fold difference in capture efficiency between HEK293T and HeLa cell derived viruses (Fig. 4), and blocking CD44 on HLFs results in ~5-fold reduction in capture and transfer, these findings are not recapitulated with FRCs (cell type most relevant to the argument put forward by the authors), where there is only 2-2.5 fold difference in capture between HeLa and HEK293T derived virus particles (Supplementary figure 3A). Additionally, pre-treatment of FRCs with anti-CD44 antibodies only resulted in a ~2-fold decrease in virus capture. In contrast to HeLa-derived virus, capture of HLAC-derived virus was only modestly inhibited by HA treatment, or blocking CD44 function on HLFs (Supplementary figure 4). These findings should be included in the main text of the manuscript and the caveats of these findings discussed.

[Response] As for the difference between HLFs and FRCs isolated from tonsils, we would like to first point out that both are equally relevant for our current model in which FRCs promote HIV-1 infection of T cells via CD44- and HA-dependent trans-infection. We confirmed HLFs, isolated from inguinal lymph nodes (ILN), are indeed FRCs based on expression of established surface markers (Fig. 1A), which were also used in this study for identification of FRCs in tonsil-derived stromal cells. As the reviewer pointed out, tonsillar FRCs show a smaller difference in capture efficiency between 293T- and HeLa-derived viruses in comparison to the HLFs, i.e., ILN FRCs (2.5-3 fold versus 5 fold), and anti-CD44 treatment is less efficient in blocking virus capture by tonsillar FRCs than that by ILN FRCs (2-2.5 fold versus 5 fold). These differences may be due to an unknown difference between FRCs from different anatomical locations. For example, it is possible that tonsillar FRCs may express another molecule(s) promoting virus capture other than CD44.

As for the difference between HLAC-derived and HeLa-derived viruses, the host molecule profile on HIV-1 particles has been known to vary depending on producer cell types [3]. Thus, it is possible that HLAC-derived virus might incorporate not only CD44 but also different factor(s) that promote capture of the virus by HLFs.

These possibilities are now discussed in the Discussion section (lines 372-388). Additionally, as requested, Supplementary Fig. 3A, Supplementary Fig. 4F, and I-M have been included as main figures (Fig. 5A, 6I, 6N and 7A-D in the revised manuscript).

3. Note that CD44 is expressed ubiquitously in most cells in vivo. HeLa cells express CD44 at levels similar to that of HLFs, based on FACS profiles presented in Fig 5 and supplementary figure 4. But the capture efficiency of HIV-1 particles by HeLa cells is significantly lower than that observed with HLFs (Fig. 1E). And, HeLa cells fail to transfer HIV to T cells. What is the basis for the cell-type specificity for CD44-mediated

capture and transfer of HIV? Do HLFs (or FRCs) express additional cell surface molecules that are necessary for the virus capture and transfer mechanism?

[Response] CD44 is a transmembrane glycoprotein that are differentially glycosylated depending on the cell types [4]. In addition, many CD44 variant isoforms, which contain one or more variant exons, exist and are differentially expressed among cell types [5, 6]. Therefore, it is quite possible that FRCs and HeLa express differentially glycosylated CD44 and/or different CD44 isoforms, which differ in virus capture efficiencies. Indeed, the differences in glycosylation and splicing patterns affect HA binding ability of CD44 [7, 8]. It is also possible that an FRC-specific cofactor may be necessary for efficient virus capture, as the reviewer suggested. These points are now included in the Discussion section (lines 389-398).

4. Similar to the capture of HIV-1 by Siglec1 (mediated by incorporation of host glycosphingolipids), HLF-mediated virus capture was dependent on incorporation of HA-bound host CD44 molecules. This is an Env-independent virus capture mechanism, but surprisingly some virus isolates (TRJO and SUMA) were captured with higher efficiency than NL4-3 (Fig 4B). What is the reason for this difference?

[Response] As the reviewer pointed out, HLFs capture some Transmitted/Founder (T/F) strains with up to 2 fold higher efficiency than other strains. Based on our previous studies, it is likely that CD44 incorporation into virions depends on the MA sequence of Gag [9, 10]. Therefore, it is conceivable that CD44 amount incorporated into virus particles could be different between virus strains due to changes in the MA sequences, resulting in the observed difference in the efficiency of virus capture. It is also possible that stability of virus particles, especially upon capture by HLFs, may differ among the strains due to the difference in the Gag sequence. At this time, however, we do not have data suggesting these possibilities, and therefore, we feel it premature to discuss these points.

5. HA is expressed as both low molecular weight (LMW) and high molecular weight (HMW) forms with differing abilities to cluster cell surface receptors, such as CD44, with differing consequences on downstream signaling. Does the type (LMW or HMW) of HA conjugated to CD44 on virus particles impact FRC/HLF mediated trans infection?

[Response] While this is an interesting point, we do not know currently which type of HA binds to CD44 on virions.

6. Previous findings on cell-associated HIV transfer, mediated by DC-SIGN, Siglec1 or mucosal stromal fibroblasts (unknown receptor), have suggested requirement for retention of captured virus at the cell surface. What is the mechanism of HLF (or FRC) mediated HIV trans infection? Does captured virus remain externalized on the HLF (FRC) surface? CD44 is a type I transmembrane protein which can internalize extracellular HA. Does virus captured by CD44 expressed on HLFs (FRCs) endocytosed? If not, does virus particle retention at the cell surface critical for HLF (FRC)-mediated

trans infection?

[Response] To test whether captured virus remains on the HLF surface, HLFs pulsed with HeLa-derived virus were treated with hyaluronidase (H-ase) for 1 hr at 37 °C. The cells were washed twice with PBS to remove unbound virus and lysed. The lysate was used for p24 ELISA to determine the amount of surface-remaining virus. We found that approximately 70% of HLF-captured virus was removed by H-ase treatment, suggesting that at least 70% of virus remains on the surface of cells. Interestingly, H-ase treatment reduced only 50% of trans-infection mediated by HLFs. Thus, it is possible that both virus remaining on the surface of HLFs and internalized virus contributes to trans-infection mediated by HLFs. Since we feel that these data should be accompanied by more mechanistic analyses, we have only discussed the possibility described above in the main text (lines 362-371), but if the reviewer thinks it essential, we are prepared to include the data as well.

7. The authors utilize FRCs and HLACs from multiple donor tonsillectomies for their studies (Fig 2). But whether HLFs were derived from multiple independent donors is not clear. The authors should provide information on whether HLFs (bought from a commercial vendor) was repeated with multiple independent stocks of HLFs.

[Response] We have used three different stocks of HLFs (3 independent donors) in this study. We have clarified in the figure legends the number of independent stocks of HLFs used for each experiment.

Reviewer #2 (Remarks to the Author):

The manuscript by Murakami and colleagues describes the impact of CD44-HA interaction incorporated in viral particles with cell surface CD44 of FRCs, and the subsequent enhanced CD44-dependent trans-infection of CD4⁺ T cells by these FRCs. While trans-infection by mucosal stromal fibroblasts was recently found, the current results are important and novel, because not only the mechanism for trans-infection (different from the mucosal fibroblasts) is now completely explored and explained, also the fibroblastic cell source is different, being FRCs from secondary lymphoid organs.

The strength of the manuscript is that with many different techniques, ie. CD44 blocking antibodies, H-ase, CD44 overexpression and CRISPR/Cas9-mediated deletion of CD44, and with different sources of primary human FRCs, the role for CD44-HA interaction and subsequent CD44-mediated virus capture and trans-infection has been very systematically demonstrated. In addition, the mechanism was also proven to work for at least 9 primary/founder HIV-1 clones. As such, the findings are important and demonstrate a new mechanism through which HIV-1 could spread in vivo in humans, clearly influencing current therapies.

Notably, however, to elevate the message even more, it would be nice to demonstrate the source (real) of CD44-HA virus particles in vivo (or ex vivo), which was suggested to be CD4⁺ T cells in the HLAC fraction from tonsils (line 261), but HA could also come from FRCs (Supplemental Figure 4D). In addition, while A3.01 seems to be a nice model for CD4⁺ T cells, the results would greatly benefit from the use of primary CD4⁺ T cells instead, as A3.01 is a T cell acute lymphoblastic leukemia. The choice for A3.01 should at least be discussed, and some experiments (eg. experiments such as Figure 1C-G) could be performed with primary human CD4⁺ T cells as well (see Neidleman et al, 2017). Lastly, while the mechanism is very nicely untangled, it would be interesting to demonstrate the trans-infection efficiency of FRCs as compared to (or together with) DCs. This would strengthen the important in vivo role of FRCs (compared to DCs) in secondary lymphoid organs in humans in the spread of HIV-1 even more. This latter point is interesting as DCs and FRCs, both interacting with T cells, also closely contact each other in secondary lymphoid organs.

How does this all fit in the model proposed in Supplementary Figure 5?

[Response] We thank the reviewer for the helpful comments. As the reviewer pointed out, since FRCs express HA on the surface, it is possible that HA bound to virus via CD44 may come from FRCs or other cells present in lymphoid organs rather than virus-producing cells. While we observed that hyaluronidase (H-ase) treatment of HeLa-derived virus reduced virus capture, in these experiments, H-ase was not removed from the virus inoculum. Therefore, in these experiments (Fig. 5H-L in the original submission; Fig. 6H and J-M in the revised version), H-ase could also degrade HA on the HLF surface. However, we also directly examined the role of HA on the HLF surface by treating HLFs, but not virus, with H-ase (original Fig. 5H). In this

experiment, H-ase was removed by washing cells prior to addition of virus. Under this condition, H-ase did not reduce virus capture by HLFs. Therefore, at least in our experimental system, it is likely that the source of HA is virus producer cells. These points are now clarified in the Results section (lines 270-280) and the legend for Figure 6 in the revised manuscript, and the detailed procedure of these experiments are included in the Methods section. Please also see our response to Major Comment below.

To address the reviewers' concern over the use of A3.01 T cell line as the target cells, we tested whether FRCs mediate trans-infection of primary CD4⁺ T cells. In these experiments, we used PHA-stimulated peripheral blood leukocytes (PBLs) as target cells for trans-infection assay. Compared to cell-free infection, the Gag-expressing cells were increased when PBLs were cocultured with virus-pulsed HLFs, suggesting that FRCs mediate trans-infection of primary CD4⁺ T cells. We added these data as new Fig. 2A and B. As requested, we also discussed the rationale for using A3.01 as target cells in our standard trans-infection assay in the Result section (lines 157-161). Please also see our response to Reviewer #1's comment.

We also compared trans-infection efficiency of HLFs with immature and mature monocyte-derived DCs. Under our experimental condition, HLFs mediate trans-infection more efficiently than immature and mature DCs. We have included these data as new Supplementary Fig. 2A and B. As the reviewer noted, DCs are in contact with FRCs in secondary lymphoid organs, which may affect the ability of DCs to trans-infect T cells or modulate FRC-mediated trans-infection of T cells. These are very interesting possibilities. However, since we have not tested whether DCs mediate trans-infection in the presence of FRCs, we feel that we are not able to provide useful discussion about the role of DCs in the context of our model shown in Supplementary Figure 6 (Supplementary Figure 5 in the original submission).

Major comment:

What is the source of HA? Although the results show that HA CD44 is produced by viral particle producing cells (mostly HeLa, Figure 5), there is only indirect/circumstantial evidence that CD4⁺ T cells (A3.01) can produce it themselves. As such, the authors state that it is likely CD4⁺ T cells isolated from tonsils (based on Figure 2). This is unfortunate, as it is important for a complete model to identify the HA producers (primary CD4⁺ T cells, FRCs, or both). Figure 5H implies that HA on FRCs inhibits capture efficiency, as it is increased upon H-ase treatment of HLFs. Yet, it seems counterintuitive (and weakening the message) that this is really the case in vivo. It seems more logical that CD4⁺ T cells produce viral particles with CD44 (either or not) lacking HA, and that FRCs capture these particles with HA-CD44 on their surface to subsequently infect uninfected T cells. To better understanding, the method of H-ase treatment should be described more extensively. Details of washing steps are missing, and also the difference between treatment of cells versus virus. Also, when was 25 units/mL H ase used (line 516)? Could experiments in Figure 1C/D be combined with H-ase treatment (and primary CD4⁺ T cells)?

[Response] Activated T cells have been shown to synthesize and express HA on the cell surface ([11, 12]; we now cite this study). Based on this information and on our data showing that inhibition of virus capture by HLFs (i.e., inguinal lymph node FRCs) occurs when virus, but not HLFs, is treated with H-ase, we think it more likely that virus producer cells including T cells are the source of HA. While we cannot rule out the possibility that HA on the FRC surface participates in capture of virus in vivo, we have no data supporting this alternative possibility at this point. On the contrary and as noted by the reviewer, H-ase treatment of HLFs enhanced virus capture approximately 1.4 fold (original Fig. 5). We interpreted these data as indicating that a subset of, but not all, CD44 molecules on HLF surface are occupied by HA and unavailable for virus capture and that H-ase treatment of HLFs remove HA from this subset, thereby making it available for virus capture. We think that this is consistent with and supporting our model rather than weakening it.

As the reviewer alluded to, his/her concern about the source of HA has likely arisen because of our failure to provide details of experimental procedures used in original Fig. 5H. We apologize for this shortcoming. In these experiments, after H-ase treatment, HLFs were washed with PBS twice to remove H-ase. Subsequently, HeLa-derived virus was added to the treated HLFs. On the other hands, when virus was treated with H-ase, H-ase was not removed before addition of the treated virus to HLFs (original Fig. 5H-L; Fig. 6H and J-M in the revised version). When we used 100 ng p24 HeLa-derived virus, treatment of virus was done with 25 units/mL H-ase. We rewrote this part of the Methods section and provided the detailed procedure of these experiments in the revised manuscript (lines 636-641).

As for the reviewer's question regarding the effect of H-ase in the T cell-HLF coculture experiments in Figure 1C/D, H-ase would degrade HA on both primary CD4⁺ T cells and HLFs. As mentioned in the Discussion section of the original and revised manuscripts, it has been reported that H-ase treatment of T cells promotes HIV-1 infection [12]. Therefore, since the interpretation of results would be difficult (T cells become more susceptible to infection, while HLFs lose its ability to mediate trans-infection), we did not perform the experiments in which the effect of H-ase on virus spread in cocultures is assessed.

Minor comments:

1. Figure 1B: it is somewhat unclear from the legend that analysis for p24 was performed on culture medium. This is always in the presence of p24 of HIV-1 (NL4-3), and either or not in presence of A3.01 T cells. Legend should more clearly state what is shown in the graph without deducting it from the experimental procedure. Also, are the adherent cells detached for p24 analysis (and replaced afterwards)? How much culture medium was collected and what did it contain (cells or medium only, or both)?

[Response] We apologize for the insufficient description of the experiments in the legend for Figure 1B. In these experiments, the adherent cells were not detached but kept in the culture during the experimental period. We collected 50 μ l of culture supernatants, i.e., medium only, for p24 ELISA analysis to monitor the amount of virus released into the medium. The remaining medium (~1 ml) was gently pipetted to make cell suspension, and 700 μ l of the cell suspension was discarded. To maintain the

cultures, we added 750 μ l of fresh RPMI1640 containing 10% FBS to each well. This information is now included in the figure legend.

2. Figure 2A-B: Virus obtained from HLAC was used to study trans-infection by HeLa/HLFs/FRCs, but what are the target cells? Primary CD4 or A3.01 T cells? This is not clear from any resource. Why is “donor-matched” HLACs important (or not)?

[Response] We thank the reviewer for pointing out this oversight. The target cells used in these experiments were A3.01 cells. We have included this information in the figure legend. To mimic the physiological condition as much as possible regarding the source of viruses, we used “donor-matched” HLAC-derived virus. However, we did not find any major difference between the “donor-matched” trans-infection and trans-infection of HLAC-derived virus by HLFs of different donors. Therefore, matching of donors between virus-producer cells and FRCs are unlikely to be important.

3. Line 140: what is the evidence in Figure 1 and 2 that demonstrate that FRCs mediate trans-infection of HIV-1 produced by T cells present in SLOs?

[Response] Figure 2A and B in the original manuscript (Figure 3A and B in the revised manuscript) provide the evidence demonstrating that FRCs mediate trans-infection of HIV-1 produced by T cells present in SLOs. In these trans-infection experiments, virus stocks were prepared by infection of HLACs. HLACs are lymphocyte mixtures prepared from tonsils, one of SLOs [13]. Since HIV-1 is likely derived from T cells in HLACs, we concluded that FRCs mediate trans-infection of HIV-1 produced by T cells present in SLOs.

4. Figure 4/Supplemental Figure 3: why was a P2 clone of A3.01 T cells used instead of parental? What is the difference of P2?

[Response] The P2 clone tends to show a polarized morphology more frequently than the parental A3.01 cells. Since the polarized morphology resembles that of T cells migrating along the FRC track in lymph nodes [14], we initially used P2 for detection of T cell-iHLF contacts in the 3D culture system. However, we later found that A3.01 T cells are similarly capable of forming contacts with the iHLF networks in the 3D culture system. Therefore, in the revised manuscript, we replaced the data obtained with P2 with those obtained with A3.01 (new Supplementary Fig. 4F).

5. Figure 4F: what type of T cells were infected and measured? Is also unclear from procedure.

[Response] Infection among CD3⁺ T cells was analyzed in these experiments. We have added this information in the figure legend.

6. Supplemental Figure 3C, D and H: significant differences?

[Response] The differences observed between HeLa- and 293T-derived viruses in Supplementary Figure 3C and D were statistically significant differences. This is now clarified in the revised figures (Supplementary Figure 4C and D). The experiments shown in original Supplementary Figure 3H (revised Supplementary Figure 4I) were performed just to confirm that two sets of 293T-derived virus stocks used in original Figure 4F (Figure 5F in the revised manuscript) are at least as infectious as corresponding HeLa-derived virus stocks used in the same experiments. While it turned out that for the two different sets of virus stocks, 293T-derived stocks were actually more infectious than HeLa-derived stocks in the TZM-bl assay, we did not use third pair of stocks of virus for Figure 4F. Therefore, statistical significance of the higher infectivity observed with the 293T-derived stocks is unknown.

7. Figure 5D, G and J: what are the target cells for trans-infection?

[Response] We used A3.01 T cells as the target cells in these experiments. We have included this information in the revised figure legend.

8. Many details such as time of inoculation, co culture etc. are mentioned in the experimental procedures (and Figure 1 only), however, some details should also be mentioned in the legends for clarity and convenience of the reader without going back-and-forth to the methods section.

[Response] To address this request and to improve the clarity, we have included some essential details of the method in the revised figure legends for new Figures 1, 3, 6 and 7.

Suggestion for the title: start with “Secondary...”

[Response] We have changed the title to “Secondary lymphoid organ fibroblastic reticular cells mediate trans-infection of HIV-1 via CD44-hyaluronan interactions”.

Line 139: add “T” to “CD4+ cells”

Line 178: T cells zone = T cell zone

Line 218: play = plays

Line 224: particle = particles

Line 235: a HA = an HA

Line 825: Date = Data

[Response] We thank the reviewer for careful reading. We have corrected these errors in the revised manuscript.

Line 834: in C. add “of A3.01 T cells” (this is the case throughout manuscript as mentioned before)

[Response] We have included this wherever appropriate.

Reviewer #3 (Remarks to the Author):

Murakami and colleagues present data demonstrating that a stromal cell found in secondary lymphoid organs (SLO), the fibroblastic reticular cell (FRC) is capable of enhancing HIV infection of Tcells through a “trans-infection” mechanism that depends on the binding of virions through interactions between the ECM receptor CD44 present on virions and on the surface of both FRC and target Tcells using hyaluronan as a bridging ligand. These facilitating interactions may be particularly important since SLO are key sites for HIV dissemination and production. The magnitude of the enhancement effects are at first seemingly modest (2 to 6 fold) but it is of interest that the % infected cells in some of the models presented are quite impressive with up to 60% of target cells infected. The paper is well written and the data are carefully presented and shed light on a novel cell-cell interaction that could have particular relevance for viral dissemination and propagation *in vivo*.

These results and the resulting model should be of interest to the journal readership. However, some additional information would increase the value of these observations to the field.

The experimental models are based on use of the lymphocytic cell line, A3.01 and HLAC (human lymphoid aggregate cells) derived from healthy donor tonsillar tissue as target cells for HIV. There is value in the use of HLAC cells since they reflect a primary cell population that seems like it would be relevant to this trans-infection system. At the same time, it would be valuable to know whether primary CD4⁺-Tcells such as those in peripheral blood would yield similar results so as to place these results in the context of the large experience and voluminous observations with *in vitro* models of primary CD4⁺Tcell infection. Related to this, the HLAC cells would presumably be a mix of many different cell types including CD8 Tcells, B cells etc. Have the authors found it necessary that all these different cell types be present in order for to see the enhancement of effect of CD44 expressing FRC? Would isolated CD4⁺TCells from the HLAC mix show the same effect?

[Response] We thank the reviewer for the helpful comments. In the original manuscript, we tested trans-infection of HLAC-derived viruses and examined virus spread in tonsil cubes, approximately 98% of which are HLACs (the rest being stromal cells), but did not specifically test HLACs as target cells in the trans-infection assay (we now clarify our description of experiments, especially regarding which cells are used as target). However, we agree with this reviewer that it would be valuable to examine whether FRCs mediate trans-infection when peripheral blood T cells are target cells. Therefore, in this revised manuscript, we investigated trans-infection mediated by FRCs using PHA-stimulated peripheral blood leukocytes (PBLs) as target cells. We observed that FRCs mediate trans-infection of CD3⁺ T cells in PBLs and enhance productive infection at 4 days post-infection. Considering the difference in the amounts of virus present in the cultures (~50 fold more present in the cell-free infection culture versus the HLF-PBL coculture), the data suggest that FRCs strongly enhance productive infection of primary T cells compared to cell-free infection. Please also see our

response to the Reviewer #1's comment above. We added the data as new Fig. 2A and B and described our findings that FRCs mediate trans-infection of primary CD4⁺ T cells in the Results section (lines 138-161). As for the potential contribution of non-CD4⁺ T cells, although this is a topic we would definitely like to pursue, because we did not test HLACs as target cells in the coculture, we have no experimental basis to discuss whether other cells affect FRC-mediated trans-infection.

In addition to the laboratory adapted NL4-3 and env- NL43 derivative, the authors studied a panel of primary, "transmitter-founder" viral clones and found similar enhancement effects for trans-infection. These transmitter-founder clones may have unique characteristics not found in primary viral isolates from those with established chronic infection. Did the authors have an opportunity to test other, "non-transmitter-founder" viral clones for comparison? Was the exclusive use of the transmitter founder panel because the trans-infection enhancement was unique to these viruses?

[Response] To address this point, we investigated whether a "non-transmitter-founder" virus strain MJ4 shows similar tendency in virus capture by HLFs. Similarly to the lab-adapted strain NL4-3 and "transmitter-founder" virus strains, HeLa-derived MJ4 virus was captured by HLFs more efficiently than 293T-derived virus. We have added the data as a new Supplementary Fig. 4A and described this finding in the Results section (lines 197-200).

In Figure Supp 4H, exogenous HA is shown to reduce infection efficiency. The HA treatment conditions are not described in the figure legend or methods and should be. Additionally, Fetal calf serum contains considerable amounts of HA and appears to have been an additive to all (or nearly all) the culture media. Can the authors address how this might have influenced experimental results in this and other experiments? Both the HA sizes added in this experiment are relatively small (<10kd and 100 to 150kd). Did the authors have an opportunity to determine whether High molecular weight HA would have a comparable inhibitory effect?

[Response] We apologize for the oversight regarding the description of HA treatment conditions. We have included this information in the Methods section (lines 641-643) and the legend for Figure 7C and Supplementary Figure 5G and H in the revised manuscript.

As for the effect of HA present in fetal calf serum, to directly address this question, we compared virus capture in the presence and absence of 10% fetal calf serum in medium during incubation of FRCs with virus. We did not see any effect on virus capture due to the presence of the serum under the condition used. This information is included in the Methods section (lines 625-628).

To address the reviewer's question about the high molecular weight HA, we compared low molecular weight HA (<10 kDa) with high molecular weight HA (>1.8 MDa) in virus capture and trans-infection mediated by HLFs. Both low molecular weight and high molecular weight HAs inhibited virus capture and trans-infection to similar extent. However, under our experimental condition, the presence of high molecular weight HA caused gelation of medium. Thus, it is unclear whether high

molecular weight HA inhibits trans-infection by blocking CD44-HA interactions or whether high viscosity of the medium prevented virus from reaching to the HLF surface. Therefore, we feel that these data may not contribute to better understanding of the effect of exogenously added HA. If the reviewer thinks otherwise, we would be happy to include the data.

1. Meran, S., et al., *Hyaluronan facilitates transforming growth factor-beta1-mediated fibroblast proliferation*. J Biol Chem, 2008. **283**(10): p. 6530-45.
2. Meran, S., et al., *Hyaluronan facilitates transforming growth factor-beta1-dependent proliferation via CD44 and epidermal growth factor receptor interaction*. J Biol Chem, 2011. **286**(20): p. 17618-30.
3. Bastiani, L., et al., *Host cell-dependent alterations in envelope components of human immunodeficiency virus type 1 virions*. J Virol, 1997. **71**(5): p. 3444-50.
4. Borland, G., J.A. Ross, and K. Guy, *Forms and functions of CD44*. Immunology, 1998. **93**(2): p. 139-48.
5. Naor, D., et al., *Involvement of CD44, a molecule with a thousand faces, in cancer dissemination*. Semin Cancer Biol, 2008. **18**(4): p. 260-7.
6. Zoller, M., *CD44, Hyaluronan, the Hematopoietic Stem Cell, and Leukemia-Initiating Cells*. Front Immunol, 2015. **6**: p. 235.
7. Bennett, K.L., et al., *Regulation of CD44 binding to hyaluronan by glycosylation of variably spliced exons*. J Cell Biol, 1995. **131**(6 Pt 1): p. 1623-33.
8. Katoh, S., et al., *Glycosylation of CD44 negatively regulates its recognition of hyaluronan*. J Exp Med, 1995. **182**(2): p. 419-29.
9. Llewellyn, G.N., et al., *HIV-1 Gag associates with specific uropod-directed microdomains in a manner dependent on its MA highly basic region*. J Virol, 2013. **87**(11): p. 6441-54.
10. Grover, J.R., S.L. Veatch, and A. Ono, *Basic motifs target PSGL-1, CD43, and CD44 to plasma membrane sites where HIV-1 assembles*. J Virol, 2015. **89**(1): p. 454-67.
11. Mahaffey, C.L. and M.E. Mummert, *Hyaluronan synthesis is required for IL-2-mediated T cell proliferation*. J Immunol, 2007. **179**(12): p. 8191-9.
12. Li, P., et al., *Exogenous and endogenous hyaluronic acid reduces HIV infection of CD4(+) T cells*. Immunol Cell Biol, 2014. **92**(9): p. 770-80.
13. Audige, A., et al., *HIV-1 does not provoke alteration of cytokine gene expression in lymphoid tissue after acute infection ex vivo*. J Immunol, 2004. **172**(4): p. 2687-96.
14. Katakai, T. and T. Kinashi, *Microenvironmental Control of High-Speed Interstitial T Cell Migration in the Lymph Node*. Front Immunol, 2016. **7**: p. 194.

Reviewers' comments:

Reviewer #1 (Remarks to the Author):

The revised manuscript by Ono and colleagues provides additional evidence for the hypothesis that secondary lymphoid tissue resident stromal cells can mediate HIV trans infection of CD4+ T cells. These are novel findings and describe a new mechanism of cell-mediated HIV trans infection. While knock-down strategies in HLF or FRCs were not provided, use of alternative anti-CD44 antibodies does provide assurance that steric hindrance was not a confounding factor. In response to the previous critiques, the authors provide additional datasets, but some of the questions remain unanswered. For instance, differences in the extent of trans infection between HeLa and FRC (both of which express CD44) is attributed to putative differences in CD44 isoforms (glycosylation differences) or requirement for a FRC-specific co-factor. The authors provide no evidence for either of these two hypotheses.

While some data with primary CD4+ T cells as target cells is presented, most of the evidence still relies on findings with A3.01 T cell line. Additionally, the small difference in cell-free and FRC-mediated trans infection of primary T cells is attributed to the large differences in virus input amount, namely 100 ng of virus used for initiating cell-free infections of CD4+ T cells (virus inoculum which was left in the culture for the duration of the experiment), while effectively, 2 ng of virus was used for initiating FRC-trans infection, because FRCs were extensively washed to remove unbound virus fraction prior to initiating co-cultures. This is not a valid comparison. While there is a clear difference in virus inputs, the fraction of infectious virus present in the 100 ng input is unclear. It's entirely possible that the 2 ng of virus captured by FRC accounts for most of the infectious virus fraction. The authors should repeat the experiment and wash away the unbound virus inoculum from the cell-free infections as well for an accurate comparison between the two modalities of infection.

Reviewer #2 (Remarks to the Author):

The authors have addressed all my concerns raised to my satisfaction. In addition, all revisions made have substantially improved the manuscript's clarity, and its message that FRCs of secondary lymphoid organs trans-infect CD4+ T cells through bridging of CD44-HA-virions on the cell surface. The study provides novel and solid findings, which are interesting to the broad readership of Nature Communications. I recommend publication of the manuscript, but still have some suggestions and minor comments for the authors that need to be addressed first, which can be found below.

Suggestions to be considered by the author:

- While clearly explained in the text, HLFs could be renamed to lnFRCs (lymph node FRCs) and FRCs to tFRCs (tonsillar FRCs), this might take away the confusion whether it are similar cell types. In this way the location of FRCs is also pointed out more clearly.

- Remove line 370-371 "More...underway." and/or show and discuss the data you have obtained, which are now stated as "(unpublished data)".
- Line 168: replace "...derived from donor-matched HLACs..." with "...derived from HLACs of the same tonsil donor..." this could take away the confusion of the importance of "matching" as done for HLA-matching in T cell experiments.
- Line 1056: replace "...we obtained similar results..." with "...similar results were obtained..." ("we" has not been used previously)
- Line 253/line 1077: are the authors using "CD44-knock out cells" or "CD44-knocked out cells" or "CD44 K.O. cells"? (perhaps use CD44-deficient cells?)

Minor comments:

- Line 165: "human lymphoid aggregated cultures" or line 439: "human lymphocyte aggregated culture"?
- Remove citations to Figures in the discussion section it does not belong there, and makes reading difficult.
- Line 1000/1001 replace "HeLa cells and HLFs or tonsillar FRCs" with "HeLa cells, HLFs or tonsillar FRCs".
- Line 1008: replace "one donor" with "a unique donor"
- Line 1008: replace "HLFs from one donor were used." With "HeLa, or HLFs from 2 donors were used."(D1/D2 = donor 1 and 2, right?)
- Line 1056: typo "form" = "from"

Reviewer #3 (Remarks to the Author):

Murakami et al report that FRC's present in secondary lymphoid organs/tissues facilitate transmission of HIV through interactions requiring hyaluronan and CD44 on both the virion and FRCs. This revision addresses most of the major issues raised in the earlier reviews and is strengthened by the inclusion of experiments confirming that infectivity is enhanced in this system when target cells are primary CD4+Tcells; by the addition of a non-transmitter founder virus experiment and by the inclusion of additional experimental/methodological detail. Overall, the paper presents an interesting, novel observation concerning the exploitation of stromal cells in SLO by HIV to promote infection and viral propagation that is well presented and justified by the data.

There are still some minor issues that the authors should review and consider amending for clarity and accuracy.

1. The y axis label in 2A and 3A read: "% of Gag+ Tcells" -- I believe they mean "% of Tcells that are Gag+" or simply "% Gag+ Tcells" and would suggest one of these labels.
2. The authors should recheck the data used to generate Fig 1D. In Fig 1C, the mixed target infected cell populations without a transwell but with HLF show a >6 fold increase in Gag+ staining. However, in 1D, this condition is shown as less than a two-fold increase, averaged across 3 replicates, and with very short S.D. bars. If in fact the column bars in 1D show

means as stated in the figure legend, then the two replicates not shown in 1C would have to correspond to a <1 fold increase in the HLF condition to arrive at a mean of the 1.8 or so shown. Could the data in 1D represent medians of the 3 replicates? In either case, the SD seems like they should be much longer. I would suggest rechecking the data for each of the 4 conditions.

3. Methods. Line 512 describes the spinoculation procedure as taking place at 2500 rpm. It would be more helpful to know the centrifugal force in g's.

Point-by-point Response to Referees

We thank the reviewers for the helpful suggestions and valuable comments. In response, in the revised manuscript, we included new data obtained through new experiments as two new figures (Supplementary Figure 3 and 6H) and described them in the Results section, all of which we think improved the manuscript substantially. In particular, following the editor's suggestion, we performed the experiment suggested by Reviewer 1, which addresses his/her concern on differences in virus input. We are pleased to see that the results, shown as Supplementary Figure 3, support our original conclusion that FRC-mediated trans-infection of primary T cells is more efficient than cell-free infection. Our responses to each of the reviewers' comments are described below under the headings of **[Response]**. In addition, as instructed by the editor, we highlighted all changes in the text file.

Reviewer #1 (Remarks to the Author):

The revised manuscript by Ono and colleagues provides additional evidence for the hypothesis that secondary lymphoid tissue resident stromal cells can mediate HIV trans infection of CD4+ T cells. These are novel findings and describe a new mechanism of cell-mediated HIV trans infection. While knock-down strategies in HLF or FRCs were not provided, use of alternative anti-CD44 antibodies does provide assurance that steric hindrance was not a confounding factor. In response to the previous critiques, the authors provide additional datasets, but some of the questions remain unanswered. For instance, differences in the extent of trans infection between HeLa and FRC (both of which express CD44) is attributed to putative differences in CD44 isoforms (glycosylation differences) or requirement for a FRC-specific co-factor. The authors provide no evidence for either of these two hypotheses.

[Response] We appreciate the reviewer for the helpful comments. The putative differences in CD44 isoforms mentioned in the Discussion section in the previous version of the manuscript are based on the literature. However, to satisfy the reviewer and to explore whether there is a difference in CD44 expressed by FRCs and HeLa cells, we performed an immunoblotting analysis of CD44 using whole cell lysates. We observed a difference in the apparent molecular weight of CD44 between FRCs and HeLa cells, an observation consistent with the possibility that FRCs and HeLa cells express different isoforms of CD44 and/or differently glycosylated forms of CD44. We added these data as a new figure (Supplementary Fig. 6H), described the result in the Results section (lines 287-291), and modified Discussion accordingly (lines 400-402).

While some data with primary CD4+ T cells as target cells is presented, most of the evidence still relies on findings with A3.01 T cell line. Additionally, the small difference in cell-free and FRC-mediated trans infection of primary T cells is attributed to the large differences in virus input amount, namely 100 ng of virus used for initiating cell-free infections of CD4+ T cells (virus inoculum which was left in the culture for the duration of the experiment), while effectively, 2 ng of virus was used for initiating FRC-trans

infection, because FRCs were extensively washed to remove unbound virus fraction prior to initiating co-cultures. This is not a valid comparison. While there is a clear difference in virus inputs, the fraction of infectious virus present in the 100 ng input is unclear. It's entirely possible that the 2 ng of virus captured by FRC accounts for most of the infectious virus fraction. The authors should repeat the experiment and wash away the unbound virus inoculum from the cell-free infections as well for an accurate comparison between the two modalities of infection.

[Response] To satisfy the reviewer's request on the amount of virus inoculum used in the PBL infection experiments, we compared the efficiency of cell-free HIV-1 infection when unbound virus is washed away with that of trans-infection. In these new experiments, we inoculated PBLs and FRCs with the same amount of HIV-1 (100 ng p24). After 2 h, both cells were washed extensively to remove unbound virus inoculum, and to FRCs, fresh PBLs were added. Following the 4-day culture, % infected T cells was determined. We observed on average 6.9-fold more infected CD3⁺ T cells in PBL-FRC cocultures than in cell-free PBL infection cultures. The results obtained with this "wash-away" protocol are now shown as Supplementary Figure 3 and described in Results (lines 154-158). In parallel with the wash-away experiments, we repeated the "no-wash" experiments (shown in Figure 2B) using the same PBL and FRC preparations. The data from these repeats are also included in Figure 2B.

Reviewer #2 (Remarks to the Author):

The authors have addressed all my concerns raised to my satisfaction. In addition, all revisions made have substantially improved the manuscript's clarity, and its message that FRCs of secondary lymphoid organs trans-infect CD4+ T cells through bridging of CD44-HA-virions on the cell surface. The study provides novel and solid findings, which are interesting to the broad readership of Nature Communications. I recommend publication of the manuscript, but still have some suggestions and minor comments for the authors that need to be addressed first, which can be found below.

Suggestions to be considered by the author:

- While clearly explained in the text, HLFs could be renamed to lnFRCs (lymph node FRCs) and FRCs to tFRCs (tonsillar FRCs), this might take away the confusion whether it are similar cell types. In this way the location of FRCs is also pointed out more clearly.

[Response] We thank the reviewer for the helpful suggestions. We have renamed HLFs and tonsillar FRCs to lnFRCs and tFRCs, respectively, in the revised manuscript.

- Remove line 370-371 “More...underway.” and/or show and discuss the data you have obtained, which are now stated as “(unpublished data)”.

[Response] We have removed the sentence in the revised manuscript.

- Line 168: replace “...derived from donor-matched HLACs...” with “...derived from HLACs of the same tonsil donor...” this could take away the confusion of the importance of “matching” as done for HLA-matching in T cell experiments.

[Response] We have replaced the sentence in the revised manuscript according to the reviewer’s suggestion (lines 173 and 1045).

- Line 1056: replace “...we obtained similar results...” with “...similar results were obtained...” (“we” has not been used previously)

[Response] We have replaced the sentence in the revised manuscript according to the reviewer’s suggestion (lines 1090-1092).

- Line 253/line 1077: are the authors using “CD44-kock out cells” or “CD44-knocked out cells” or “CD44 K.O. cells”? (perhaps use CD44-deficient cells?)

[Response] We now use “CD44-knocked out cells” in all applicable cases in the revised manuscript (lines 258, 260, 512-514, 550, and 1112-1113).

Minor comments:

- Line 165: “human lymphoid aggregated cultures” or line 439: “human lymphocyte aggregated culture”?

[Response] Both terms appear to be used in the field. In this manuscript, we have chosen “human lymphoid aggregate cultures” and used it consistently in the revised manuscript (lines 170 and 1034).

- Remove citations to Figures in the discussion session it does not belong there, and makes reading difficult.

[Response] We have removed some of citations to Figures in the discussion. However, to clarify the experimental basis of our discussion, we still have a few citations to figures in the revised manuscript.

- Line 1000/1001 replace “HeLa cells and HLFs or tonsillar FRCs” with “HeLa cells, HLFs or tonsillar FRCs”.

[Response] We have replaced the sentence as suggested in the revised manuscript (line 1035).

- Line 1008: replace “one donor” with “a unique donor”

- Line 1008: replace “HLFs from one donor were used.” With “HeLa, or HLFs from 2 donors were used.”(D1/D2 = donor 1 and 2, right?)

[Response] We apologize for the mistake and thank the reviewer for finding it. We have replaced the wordings as suggested in the revised manuscript (line 1042).

- Line 1056: typo “form” = “from”

[Response] We apologize for the typo. We have corrected this error in the revised manuscript (line 1060).

Reviewer #3 (Remarks to the Author):

Murakami et al report that FRC's present in secondary lymphoid organs/tissues facilitate transmission of HIV through interactions requiring hyaluronan and CD44 on both the virion and FRCs. This revision addresses most of the major issues raised in the earlier reviews and is strengthened by the inclusion of experiments confirming that infectivity is enhanced in this system when target cells are primary CD4⁺Tcells; by the addition of a non-transmitter founder virus experiment and by the inclusion of additional experimental/methodological detail. Overall, the paper presents an interesting, novel observation concerning the exploitation of stromal cells in SLO by HIV to promote infection and viral propagation that is well presented and justified by the data.

There are still some minor issues that the authors should review and consider amending for clarity and accuracy.

1. The y axis label in 2A and 3A read: “% of Gag⁺ Tcells” -- I believe they mean “% of Tcells that are Gag⁺” or simply “% Gag⁺ Tcells” and would suggest one of these labels.

[Response] We thank the reviewer for the helpful suggestions. We have changed the labels to % Gag⁺ T cells in the revised Figures.

2. The authors should recheck the data used to generate Fig 1D. In Fig 1C, the mixed target infected cell populations without a transwell but with HLF show a >6 fold increase in Gag⁺ staining. However, in 1D, this condition is shown as less than a two-fold increase, averaged across 3 replicates, and with very short S.D. bars. If in fact the column bars in 1D show means as stated in the figure legend, then the two replicates not shown in 1C would have to correspond to a <1 fold increase in the HLF condition to arrive at a mean of the 1.8 or so shown. Could the data in 1D represent medians of the 3 replicates? In either case, the SD seems like they should be much longer. I would suggest rechecking the data for each of the 4 conditions.

[Response] As described in the legend, Figure 1D showed the mean of the fold-increase in Gag⁺ target cells AFTER normalization by fold-increase in Gag⁺ donor cells. On the other hand, Figure 1C is a representative data set of dot plots for only target cells and hence BEFORE normalization. The contacts of donor cells with HLFs increased virus spread 3-4 fold among donor cells (Supplementary Fig. 1), which could consequentially contribute to the increase of virus spread in target cells. To avoid overestimating the enhancing effect of FRC on virus spread specifically in target cells, we normalized the fold-increase in virus infection in target cells by the fold-increase in infected donor cells. To clarify this point, we have changed the y-axis label to “Fold-increase in Gag⁺ target cells / fold-increase in Gag⁺ donor cells” in the revised Figure 1D.

3. Methods. Line 512 describes the spinoculation procedure as taking place at 2500 rpm. It would be more helpful to know the centrifugal force in g's.

[Response] We have changed “2,500 rpm” to “2,500 rpm (888 or 1,462 xg depending on the centrifuge used)” in the revised manuscript (lines 518-519).

In addition to changes requested by the editor and the reviewers, we made two additional changes in this revision.

1. We added additional information for the literature regarding distribution of FRCs within SLOs (lines 53-54).
2. We switched Figure 7C and 7D to match the order of panels to that of the text.
3. We indicated positions of molecular weight markers in gel figures (Figures 6 and Supplementary Figure 6).
4. We added Acknowledgments.